



# Atmospherically relevant chemistry and aerosol box model – ARCA box (version 1.2.0)

Petri Clusius[1*], Carlton Xavier[1*], Lukas Pichelstorfer[2], Putian Zhou[1], Tinja Olenius[3], Pontus Roldin[4], Michael Boy[1]

[1] Institute for Atmospheric and Earth Systems Research/Physics, University of Helsinki, P.O. Box 64, 00014 Helsinki, Finland
[2] Department of Chemistry and Physics of Materials, University of Salzburg, 5020, Salzburg, Austria
[3] Swedish Meteorological and Hydrological Institute (SMHI), SE-601 76 Norrköping, Sweden
[4] Division of Nuclear Physics, Department of Physics, Lund University, P. O. Box 118 SE-221 00 Lund, Sweden
* These authors contributed equally to this work.

*Correspondence to*: Petri Clusius (petri.clusius@helsinki.fi)

**Abstract.** We introduce the Atmospherically Relevant Chemistry and Aerosol box model ARCA box (v.1.2.0). It is a zero-dimensional process model with focus in atmospheric chemistry and submicron aerosol processes, including cluster formation. Novel feature in the model is its comprehensive graphical user interface, allowing for detailed configuration and documentation of the simulation settings, flexible model input and output visualization. Additionally, the graphical interface contains tools for module customization and input data acquisition. These properties – customizability, ease of implementation and repeatability – make ARCA invaluable tool for any atmospheric scientist who needs a view on the complex atmospheric aerosol processes. ARCA is based on previous models (MALTE-BOX, ADiC and ADCHEM) but the code has been fully rewritten and reviewed. The chemistry module incorporates the Master Chemical Mechanism (MCMv3.3.1) and Peroxy Radical Autoxidation Mechanism (PRAM) but can use any compatible chemistry scheme. ARCA's aerosol module couples the ACDC (Atmospheric Cluster Dynamics Code) in its particle formation module, and the discrete particle size representation includes the fully stationary and fixed grid, moving average methods. ARCA calculates the gas-particle partitioning of low-volatility organic vapours for any number of compounds included in the chemistry, and the Brownian coagulation of the particles. The model has parametrisations for vapour and particle wall losses but accepts user supplied time and size-resolved input. ARCA is written in Fortran and Python (user interface and supplementary tools), can be installed on any of the three major operating systems and is licensed under GPLv3.

## Introduction

Aerosol and chemical box models find wide use in many practical air quality or environmental questions. During the last decades of atmospheric research, several numerical process models for simulating gas and particle phase chemistry and dynamics have been developed and used to study ambient phenomena (e.g., Boy et al. 2006; Boy et al. 2011; Roldin et al. 2011; Roldin et al. 2014; Roldin et al. 2019), study complex chemical schemes (e.g., Boy et al. 2013; Xavier et al. 2019) or specialized applications such as chemistry and deposition in lungs (e.g., Pichelstorfer and Hofmann 2015; Pichelstorfer et al. 2021).

When a scientist without prior experience of a detailed process model wants to apply one, often practical problems arise. Many of the models require substantial modification of source codes and programming experience to be applicable for a given case study. Available models are usually tailor-made for a particular problem and might not be written with flexible enough code (for example, parameters are hard-coded) and consequently be vary laborious to set up. Setting up a model involves detailed configuration, usually requiring in-depth knowledge of the code structure, and contains high risk of misconfiguration. The more the model code needs to be modified, the more the probability of introducing bugs increases. These are some reasons why applying complex numerical models can be unappealing. For most scientists, the simulation



itself is not the main focus or motivation of the work, and one would prefer flexible and easily applicable tools, with minimal risk of misconfiguration, and with reasonable amount of time spent in familiarization with the tool. Recently, Python-based solutions have become available, e.g., PyCham (O'Meara et al. 2020) and PyBox (Topping et al. 2018). Yet, it can be said that in terms of usability, the scientific tools are not on par with current-generation software in general. To meet these

challenges, we introduce the 0-dimensional Atmospherically Relevant Chemistry and Aerosol box model ARCA box v1.2.0. One of the objectives of the development has been in applicability of the model. It is flexible regarding the complexity of the chosen chemistry and aerosol composition as well as the timescale of the simulation. The backbone of ARCA consists of established theories and standard model implementations, but the model is flexible to customization and further extensions. The source code is written in a way that enables the use of additional or substituting parametrizations for the modelled

processes. However, the major advantage of ARCA, setting it apart from other current models in its field, is the graphical user interface (referred from here on as GUI). It makes the model easier to apply, greatly increases reproducibility, reliability and documentation of the simulations, provides tools for visualisation of the output and automatizes many steps in model setup and configuration.

This paper is structured in the following way: Section 1 introduces the scope of the model and explains its structure and main

functionality. Section 2 describes the scientific theories behind the modules. Section 3 explains the usage of the model, focusing on the graphical user interface. Section 4 shows verification and standard evaluation of the model's modules. Section 5 summarises technical details about the system requirements, installation, licensing, code availability and further documentation (the ARCA online manual). Section 6 concludes this paper with plans for future development.

When text is written in MONOTYPE, it refers to a user-definable variable name, which are available from the GUI. Full list of

these input variables is shown in Appendix A.

## 1 Scope and uses of the ARCA box

ARCA box is primarily intended to be used for studying processes such as gas phase chemistry and aerosol processes at atmospherically relevant concentrations (trace gases) and conditions (pressure, temperature, humidity, irradiance). A box model is typically used for simulating smog chambers, indoor spaces or other small containers. Additionally, it can be used

to simulate ambient (field) processes, as long as the assumption of homogeneity of the conditions are met (see 1.1). Using a box model instead a dimensional model in outdoor simulations is beneficial because computational resources can be put in more detailed chemistry and aerosol processes. ARCA is therefore well suited for studying complex processes and developing and testing new (chemical, aerosol, etc.) schemes, before implementing them in a dimensional model. In addition to scientific research, due to its ease of use and configuration ARCA has also been used in teaching of aerosol chemistry and

physics.

Given a proper chemistry scheme the model can be used to study formation of chemical compounds from precursors (or their emissions), calculate effective reactivities (inverse of chemical lifetime) with chosen reactants, and simulate the effects of dynamically varying conditions to these processes. The particle formation rate module, containing the Atmospheric Cluster Dynamics Code (ACDC, McGrath et al. 2012; Ortega et al. 2012) simulates the production of new nanoparticles by

clustering of molecules – by default from $H_2SO_4$–$NH_3$ and $H_2SO_4$–Dimethylamine (DMA) mixtures, but any chemistry can be included, given compatible input data. Any organic compound in the chemistry scheme, whose pure liquid saturation vapour pressure is known (or estimated), can contribute to particle growth by condensation as calculated by the Analytical Prediction of Condensation (APC) scheme (Jacobson 2005). Aerosol processes further include coagulation losses and growth by Brownian coagulation, and losses by external sink such as wall losses. Because any of these processes can be switched on

or off, quantifying their effects to the total dynamics is straightforward. The GUI allows model initialization and constriction in different ways, using either through predefined values from files (such as measurements), or by parametric, time





dependent functions, configured graphically in the GUI. Sensitivity studies, used to assess the effects of uncertainties and variability of the model parameters, are done by changing the parameters within some range. To this end, the GUI has tools to create batches of simulations, where the nominal time dependent input parameters for selected variables are varied (either

by multiplying or shifting) within user defined ranges and intervals.

### 1.1 Main assumptions of the model

Like any box model, ARCA does not consider spatial variation and the related processes, most importantly advection (including convection). When simulating ambient (field) processes, we must therefore assume that the conditions are spatially homogeneous. The transport equation, describing the local change of scalar variable $S$ in terms of its total derivative

and advection in wind field $\vec{V}$ , then becomes

$$\frac{\partial S}{\partial t} = \frac{dS}{dt} - \vec{V} \nabla S \approx \frac{dS}{dt} \tag{1}$$

which is acceptable approximation when $\left|\frac{dS}{dt}\right| \gg \left|\vec{V} \nabla S\right|,$ as is the case with fast chemical reactions, but not necessarily for slow processes like aerosol growth and coagulation losses. This alone will produce inevitable deviation between modelled (following $dS/dt$ ) and locally measured (following $\partial S/\partial t$ ) time series.

In the aerosol module particles are assumed to be spherical, liquid droplets with constant density (`ORGANIC_DENSITY`).

Charges are omitted and all particles in the aerosol module are treated as electrically neutral (ion-mediated nucleation is simulated in ACDC). Liquid phase organic chemistry in the particles, for example polymerization and the consequential effects this has on the thermodynamics (sometimes called ageing) is not considered in the model. In the present model version particles of all sizes – even above activation diameter – are completely void of water and the dissolution of inorganic compounds is therefore ignored. Some of these restrictions will be addressed in the near future updates (see section 6).

The ACDC nucleation module is more flexible than the aerosol module and – given the proper input – would be capable of simulating also hydrated clusters, but presently neither of the included cluster chemistry systems contain hydrated clusters.

### 1.2 Structure of the model

The main processes modelled in ARCA box are (in the order of execution): 1) gas phase chemistry 2) formation of molecular clusters 3) reversible vapour wall loss 4) gas-particle partitioning (condensation and evaporation) 5) coagulation of particles

6) wall loss of particles (Fig. 1). The processes – which can be switched on and off in any combination – are executed in a series where the next process relies upon values which were calculated in the previous process. Compared to a method where all changes within time step are solved as one coupled system, this has the advantage that adding more processes (or skipping them) is straightforward as they can simply be added to the main loop as a subroutine or module and solved in any suitable way. On the other hand, the forward integration requires that the time step is kept small enough to justify the

linearisation of a non-linear system.

Programmatically the model consists of two main parts: the numerical model (in Fortran), which should be compiled on the computer where it is executed, and the graphical user interface GUI (in Python). Both software environments are freely available in Windows, Mac and Linux, and the model has been successfully installed and used on all three platforms. The numerical module is configured and initialized with a setting file, called hereafter as INITFILE, defining all the simulation

settings. Additional input includes the spectral irradiance or actinic flux, pure liquid saturation vapour pressures of low-volatility organic compounds and (optionally) the elemental composition of the condensing vapours, and (also optionally) time series of environmental variables, precursor gases, initial or background aerosols and aerosol loss rates, if the built-in parametrization is not used.



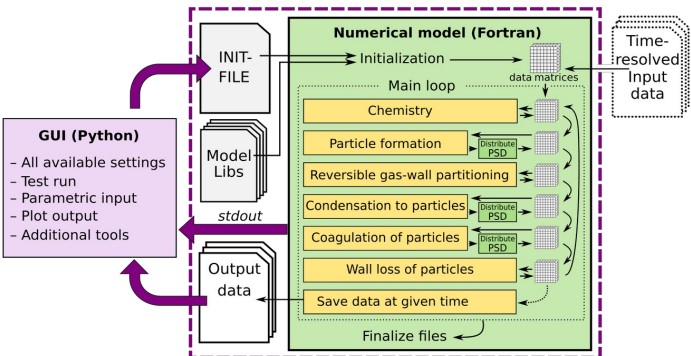

**Figure 1: Schematic representation of ARCA box. The green rectangle contains the Fortran part of the model, yellow boxes contain the main modules, the purple box the Python graphical user interface (GUI). Purple arrows show where the interaction between the GUI and Fortran executable takes place. GUI interacts with the Fortran model by writing the INITFILE (top purple arrow), repeating the screen output of the numerical model (middle purple arrow, *stdout*) and plotting the output data (bottom purple arrow). The dashed purple rectangle is the minimal configuration needed to run ARCA, Input data is not strictly necessary as parametric input can also be used.**

### 1.3 Model time step optimization

The model can be used with a fixed time step DT (default time step is 10 s) or with multiple, dynamically changing time steps. With latter the user defines the maximum and minimum relative change in one time step for 1) particle diameter ($dd_p$), 2) particle number concentration in each bin ($dN_p$) and 3) concentration of condensable vapours ($dC$). After exiting each module (shown in Fig. 1), the changes in 1–3 are calculated. If the module produced changes that exceed the maximum, the

time step for that module is halved and the current integration time step will start over. In contrast, when the largest change is below the tolerance minimum, the time step of that module is doubled. The modules are assigned with one of the three time steps, denoted CCH (condensation and chemistry), COA (coagulation) and DEP (losses). The processes used to control the time steps are shown in Table 1.

**Table 1: Time step optimization.**

| Module | Time step | Defining process |
|---|---|---|
| Chemistry | CCH | $dC$; if the Condensation module is used, chemistry does not affect time step; otherwise, $dC$ is calculated for compounds whose concentration is above MIN_CONCTOT_CC_FOR_DVAP (default 1000 molec/cm³) |
| Particle formation | min(1,2,3) | [does not affect time step]. When used without other modules, time step optimization is not available. |
| Chemical wall loss | CCH | [does not affect time step] |
| Condensation | CCH | $dC$, $dD_p$; $dC$ is calculated for compounds whose concentration is above MIN_CONCTOT_CC_FOR_DVAP |
| Coagulation | COA | $dN_p$ |
| Particle losses | DEP | $dN_p$ |

When time step optimization is used, a 3-element vector DT_UPPER_LIMIT defines the maxima for CCH, COA, and DEP, and the nominal model time is used as a minimum (by default the GUI sets DT to 1 ms). Therefore, the three time steps are





defined by multiplying the nominal model time step by a factor of $2^n$, where n = {1, 2, 3...}, and satisfy $\texttt{DT} < \Delta t \times 2^n \leq$ $\texttt{DT\_UPPER\_LIMIT}$. Optimizing time step this way has two effects. It helps to overcome the potential error created by

decoupling of different modules that are not solved together as one dynamic equation (notably condensation to particles and chemistry) by guaranteeing that the changes in one time step are small. On the other hand, when a process is very slow, such as aerosol coagulation, the extra precision gained by small time step is negligible; instead increasing the time step can shorten the computational time significantly. The effect on the execution time of the simulation depends on the tolerances and the conditions of the simulation. The time step optimization should first and foremost be seen as a safeguard against

diverging too far from the true solution of the integration.

## 2 Modules of ARCA box and their theoretical base

### 2.1 The chemistry module

The chemistry code of ARCA is based on the KPP (Kinetic PreProcessor, https://people.cs.vt.edu/asandu/Software/Kpp; Damian et al. 2002), and any reaction set which complies with the KPP format can be used to create ARCA's chemistry

modules. An often-used source for atmospheric chemistry scheme is the MCM v3.3.1 (Master Chemical Mechanism version 3.3.1, http://mcm.leeds.ac.uk/MCMv3.3.1) (Jenkin et al. 1997; Saunders et al. 2003; Jenkin et al. 2015), which provides the gas-phase chemical reactions and rates of the degradation of specific organic compounds in the atmosphere, as well as detailed inorganic chemistry and photochemistry. The full scheme – or a subset – can be downloaded from MCM website in KPP format. Optionally, the user might have additional chemistry schemes, which need to be combined with the MCM

scheme, such as the PRAM scheme (Peroxy Radical Autoxidation Mechanism; Roldin et al. 2019) which describes the production of highly oxygenated molecules (HOMs) from monoterpenes and is included in the ARCA distribution. ARCA's GUI has a tool ("Create chemistry scheme") which combines different schemes, removes duplicate compounds and reactions and creates a valid single KPP definition file, used for producing a Fortran module with KPP. A modified source code of KPP v2.2.3, accommodated for very large schemes, is provided with ARCA. The generated code will then be part of ARCA

and is available for use after compilation. Switching between different chemistry schemes can later be done in the GUI with a dropdown menu.

### 2.1.1 Photochemistry

The photolysis rates of the photochemical reactions are calculated by integrating the absorption cross section, quantum yield (provided by MCM v3.3.1), and actinic flux (AF). Actinic flux data (in W m⁻² nm⁻¹) can be sent directly in or it can be

estimated from the global short-wave radiation (also in W m⁻² nm⁻¹), surface albedo, geographic location and the date of the simulation, as described in Kylling et al. (2003). The actinic flux and short-wave radiation are wavelength dependent, but often only the total irradiance over a wavelength range is measured, instead of the spectral distribution. ARCA contains a generic clear sky spectrum (*glob_swr_distr.txt*) between 280 nm – 4 µm, obtained from the Bird Simple Spectral Model (Bird and Riordan 1986). There is also a site-specific spectrum for SMEAR II station in Hyytiälä, Finland, based on

measured and averaged yearly spectra from 2001 (Boy and Kulmala 2002). The spectrum in *glob_swr_distr.txt* is normalized for 300 nm – 4 µm wavelength range, assuming a flat response from the instrument. Should the actual measurement range differ from this, the user can provide the lower and upper limits of the wavelengths, and the default spectrum will be normalized to this range.

The provided two spectra are somewhat generic and represent field conditions. For more precision – or e.g., smog chamber

simulations – the user should provide their own spectral data which must be in 5 nm steps and contain entries from 280 nm to 700 nm. Exceeding entries in the data file are ignored as they are not relevant in the photochemistry. Spectral data can be





a single, constant spectrum or time-resolved. If time dependent data is provided, a linear interpolation in time is performed at each model time step. The exact format of the spectral file is described in the online manual.

The user supplied spectral data can either be thought as a weighing function, whose integral over the measured wavelength

range amounts to 1, or as absolute values [W m⁻² nm⁻¹]. In the first case the time-dependent scalar variable `SW_RADIATION` represents the measured total irradiance, and in the latter `SW_RADIATION` is set to a constant (1, or some dimensional factor). Finally, if the option `SWR_IS_ACTINICFLUX` is selected, the `SW_RADIATION` is directly used as actinic flux. A schematic representation of the short-wave radiation data processing is shown in Fig. 2.

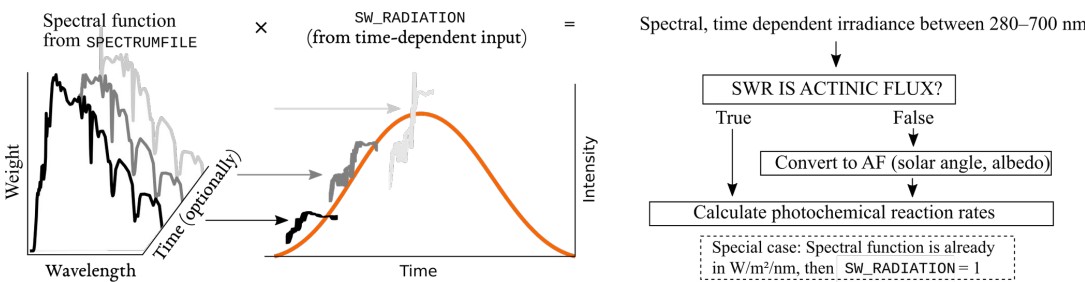

**Figure 2: Schematic representation of how actinic flux is calculated from the provided input.**

**2.1.2 Reaction kinetics**

At each time step, the reaction rates (including photolysis) are updated using current pressure, humidity, irradiation and temperature. With these the chemistry module solves the new concentrations after one model time step. The concentration $C$ of any compound $q$ is described by the ODE

$$\frac{dC_q}{dt} = \sum_{x=1}^{N_q} R_{x,q} \qquad (2)$$

where $R_{x,q}$ are the reactions $x$ that either produce or consume the compound $q$. The ODEs of the compounds in the chemistry

scheme form a system of coupled non-linear equations, which is solved numerically in ARCAs chemistry scheme, by default using KPPs general Rosenbrock solver (Sandu et al. 1997).

In ARCA the concentration of any compound in the chemistry can be set to a user defined value. Typically, this is done for the precursor gases, whose concentrations could be measured or otherwise known. The user supplied concentrations are read from the input, interpolated to model time and saved in the model data matrix, overwriting previous values. Then, if a

compound was defined as fixed in the chemistry scheme (DEFFIX in KPP definition file), its time derivative (eq. 2) is zero, and the concentrations do not change during the chemistry step; if the compound was not fixed (DEFVAR in KPP definition), the concentrations will in general change (and this would be reflected in the concentrations of its reaction products), but as previously, the resulting concentration will be overwritten in the next time step when the model concentration is set to the given input concentration. The user can also define a time after which the updating of the

concentrations of the chemical compounds (and separately for emissions) is stopped, and the chemistry is let to "float", meaning that also the precursor concentrations evolve dynamically without outside interference (if they are DEFVAR). The definition of fixed and varying compounds can easily be done in ARCAs "Create chemistry scheme" tool when creating a new chemistry scheme. As a rule, precursors should be defined as fixed, unless the simulation involves floating of the chemistry, as described above. Figure 3 further explains the effect of these settings to the simulation.



After the chemistry step, all negative values are set to zero. Negative concentrations are non-physical but could emerge from numerical inaccuracies, so a warning is issued if some concentration is less than −100 molec/cm³, as this would be an indication of misconfiguration, such as wrong units in the input.

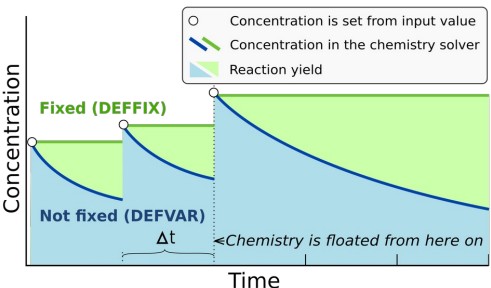

**Figure 3: Schematic example of the effect of defining a compound in DEFFIX or DEFVAR. In this example, the variable is set by the user (in the GUI, this is done in the tab Time dependent input). The solid lines show the concentrations inside the chemistry solver for a compound which is fixed (green curve), and not fixed (blue curve). The circular symbols show the set concentration in each time step. The shaded areas represent the proportional impact of DEFFIX and DEFVAR on the concentrations of the reaction products of the compound. Depending on the model time step Δt, the differences can be substantial, and illustrates why the precursors should be set fixed, unless the chemistry needs to be floated.**

### 2.2 Particle size distribution (PSD)

The main use of ARCA, besides simulating chemical reactions in the gas phase, is to compute the evolution of a population

of aerosol particles experiencing dynamical processes such as coagulation, deposition and phase transition. There are many ways to formally approach this task, discussed e.g., by Whitby and McMurry (1997). In the atmosphere, the number of condensing species, different particle sizes and compositions is unlimited, and the practical mathematical descriptions of the PSD therefore require simplifying assumptions. The aerosol dynamics models are often characterized by their representation of the PSD. Typical representation types are the discrete, the spline, the sectional, the modal and the monodisperse

representation.

In the discrete representation, particles have a discrete size and composition. That is, one size section includes exactly one discrete particle composition, given by the exact numbers of molecules of different compounds in the particle. This approach must be used when explicitly modelling initial cluster formation but gets increasingly complex and numerically heavy when more 'building blocks' are added, such as chemical compounds or particle sizes. ARCA's nucleation module consists of

ACDC model which uses discrete representation.

Sectional models decrease complexity of the system by grouping parameters, and typically the particle diameter range (and equivalently volume, mass and surface) is divided into a limited number of intervals, or size bins. Particles in each bin have the same properties. This is a common approach in atmospheric modelling, however, its accuracy depends on the number of bins used and on the magnitude of change applied within one integration time step $\Delta t$. While large changes within $\Delta t$ lead to

a deviation from the analytical solution, small changes cause numerical diffusion especially when simulating condensational growth (Gelbard and Seinfeld 1980).

ARCA allows to choose between two sectional representations, the Fully Stationary (FS; Gelbard and Seinfeld 1980) method and the fixed grid moving average method (MA, also called 'moving centre sectional'; Jacobson 1997a). Future roadmap includes a hybrid method, discussed in section 6. The current representation methods are graphically summarized in Fig. 4.



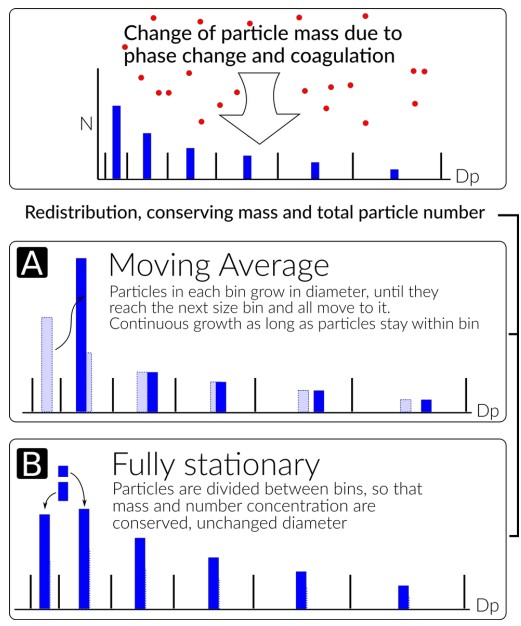

**Figure 4: Schematic of the two particle size distribution representations, the fixed grid, moving average (MA, panel A) and the fully stationary (FS, panel B).**


The FS method is very robust during computation but suffers from numerical diffusion. The MA method strongly reduces numerical diffusion as the particle diameter within the size grid can continuously change and particles are not distributed between bins. Main drawback of the MA method is the appearance of numerical artefacts, referred to as "pits" and "peaks", affecting the analysis of the results. However, they can be removed during data analysis by remapping the data on a new size grid as described by Mohs and Bowman (2011).


The choice of the PSD representation does not affect the calculations in the aerosol dynamics module as they are separated from each other. This means that the different methods to represent PSDs can be changed or added without interfering the aerosol dynamics code. Aerosol related parameters, required to solve the dynamics, are provided by accessor functions (getters) and work for any PSD representation. Further, changes to the PSD by the dynamic processes are calculated in the aerosol dynamics module but applied in the PSD module. This allows further development of the scientific and computational aspects independently.


### 2.3 Initial new particle formation by molecular clustering

Formation of new particles from vapours by clustering of gas-phase molecules, often referred to as aerosol nucleation, is described by simulation of molecular cluster population dynamics with quantum chemistry input for cluster evaporation rates. This yields the formation rate of new particles per unit volume and unit time.


#### 2.3.1 Molecular cluster dynamics simulation

Molecular cluster formation dynamics is solved by the ACDC code (https://github.com/tolenius/ACDC), which calculates the time-dependent cluster number concentrations for a given set of molecular clusters and ambient conditions. A detailed description of ACDC can be found in (Olenius et al. 2013). Briefly, ACDC generates and solves the discrete, molecular-resolution general dynamic equation, also known as the cluster birth–death equation, considering all possible collisions and evaporations among the clusters and vapour molecules, ionization and recombination processes, and external cluster sinks. When a collision results in a cluster that is outside the simulated system, and its composition satisfies the defined stability




criteria, it is considered stable enough to not re-evaporate (Olenius et al. 2013). These outgrown clusters constitute the formation rate of new particles.

By default, collision rate constants are calculated as hard-sphere collision rates for collisions between electrically neutral species, and according to dipole-moment and polarizability-dependent parametrizations for collisions between neutral and ionic species (by default from Su and Chesnavich (1982)). Evaporation rate constants are obtained from quantum chemical cluster formation free energies. Ionization and recombination of clusters and molecules occur by collisions with primary ionic species, which originate from galactic cosmic rays and radon decay and are assumed to have the properties of $O_2^-$ and

$H_3O^+$. Cluster scavenging sink is obtained from the particle population modelled by ARCA, with sink rates $R_{sink}$ calculated according to $R_{sink} = CS_{vapor} \times (D_{cluster} / D_{vapor})^{-1.6}$, where $D_{cluster}$ is cluster diameter, $CS_{vapor}$ and $D_{vapor}$ are the condensation sink and diameter of the monomer ($H_2SO_4$ is used as proxy), respectively (Lehtinen et al. 2007). $CS_{vapor}$ can also be give as time dependent input (CONDENS_SINK), if the aerosol module is not in use. The user familiar with ACDC may also modify the rate constants and the settings for the inclusion of different cluster dynamics processes. In addition to the ARCA user manual, the

interested reader is referred to the ACDC manual, available from the repository, and references therein.

### 2.3.2 Available cluster chemistries

The current default implementation includes two separate clustering chemistries: $H_2SO_4$–$NH_3$ and $H_2SO_4$–DMA (dimethylamine), including electrically neutral, negatively and positively charged clusters. Cluster evaporation rates are in the default installation calculated from previously published quantum chemical data computed at the

B3LYP/CBSB7//RICC2/aug-cc-pV(T+d)Z level of theory (Olenius et al. 2013), but the user may switch to alternative systems, calculated with DLPNO-CCSD(T)/aug-cc-pVTZ//ωB97X-D/6-31++G** level of theory. The main reasons for applying the B3LYP//RICC2 data are that these cover large sets of cluster compositions including all charging states and performs reasonably well compared to laboratory experiments (Almeida et al. 2013); Kürten et al. 2016). However, cluster evaporation tends to be underpredicted and thus the formation rates can be considered upper-limit estimates. ARCA's

procedures allow easy rebuilding of additional cluster chemistries, if the user wants to apply new or updated data for the evaporation rates. ARCA has slots for five ACDC modules, but the number can be increased with minor code modifications.

### 2.3.3 Coupling of cluster and aerosol dynamics

The cluster dynamics simulation is implemented by passing the input for the ambient conditions to the cluster formation routine at each model time step. The input includes concentrations of the clustering vapours, temperature, primary ion

production rate (if supplied by the user), and condensation sink of $H_2SO_4$ (from previous time step or CONDENS_SINK), used as reference molecular size for scavenging sink $CS_{vapor}$. The cluster formation routine solves the time evolution of the cluster concentrations for the given time step, using the concentrations from the end of the previous time step as initial values, and returns the number of new particles that grow out of the cluster regime during the time step. This is converted to new particle formation rate (by $R_{formation} = C_{out}/\Delta t$) and handed to main module. The newly formed particles are then distributed within the

first model PSD size bins in the diameter range of $[\min(d_p), 1.15 \times \min(d_p)]$, with weights calculated by

$$w_i = 0.5 \exp(-0.7 i), i = \{2 \dots d\}, w_1 = 1 - \sum_{i=2}^{d} w_i ,$$ (3)

where $d$ is the index of the bin closest to diameter $1.15 \times \min(d_p)$. This somewhat arbitrary distribution is based on the assumption that some stable clusters result from collisions that would produce larger than the minimum stable cluster, and assigns about 75–85% of the newly formed particles to the first bin. The factor to calculate the upper diameter for the

distribution (by default 1.15) can be changed with NPF_DIST (e.g., set to 1). The user is responsible for selecting a suitable PSD size range so that the minimum size for the simulation closely matches the size of the outgrowing particles from the



ACDC systems. In case the selected $\min(d_p)$ in the model is larger than outgrowing cluster from any of the ACDC subsystems, the model issues a fatal warning in the beginning of the run and terminates. If the outgrowing clusters are more than 10% larger than $\min(d_p)$, a (non-fatal) warning is issued in the first time step. The newly formed particles are assigned a

general non-evaporating particle composition (in the model called GENERAL).

ACDC has an option to calculate a steady-state formation rate, which corresponds to a time-independent situation where the cluster concentrations have relaxed to a steady state for the given ambient conditions. The steady-state option (ACDC_SOLVE_SS) must not be used for a dynamic atmospheric case where the conditions vary with time, and thus the formation rate depends on the immediate history of the conditions. Instead, steady state option is useful if the user wishes to

run the cluster routine as a stand-alone model and study the dependence of the steady-state formation rate on, for example, vapour concentration or temperature. In general, while the steady-state approximation is necessary for computationally heavier large-scale models, it may cause artefacts especially at low vapour concentrations or dynamic conditions. As an explicit cluster simulation is easily embedded in a box model such as ARCA, it is reasonable to not introduce such unnecessary potential error source.

### 2.4 Other options for new particle formation rate

In addition to the explicit cluster dynamics simulation, simplified formation rate parametrization can be used. Currently, a parametrization for new particle formation from $H_2SO_4$ and representative organic components is available. This option approximates the formation rate as a function of $H_2SO_4$ and user-defined set of organic compound concentrations. The choice of what organic compounds (or proxies for them) to include in the set is not unambiguous and is dependent on the

active chemistry scheme and assumptions made by the user. Still, such parametrizations are commonly used to characterize empirical observations of particle formation (e.g., Paasonen et al. 2010). While they do not correspond to an explicit time-dependent formation rate and may not include all independent ambient parameters which could affect the formation rate, they are useful for assessing the magnitude of particle formation through the given chemistry.

To compliment the formation rates and the parametrization, the model has also a time-dependent input variable

NUC_RATE_IN. When multiple particle formation rates are calculated, either by different ACDC systems, additional parametrizations or NUC_RATE_IN, the total formation rate – used in the aerosol module – is the sum of all processes. The output files contain the total formation rate, and separately those from the ACDC systems. Furthermore, if applicable, the formation rates of electrically neutral and positively and negatively charged particles is also saved. It must be emphasized that the charging states refer solely to the charges of the outgrowing particles in ACDC, not to the formation mechanisms of

the particles. For instance, newly formed neutral particles may originate – and often do originate (Olenius et al. 2013) – from ion-mediated processes, where small ions recombine and the product grows further as a neutral cluster.

### 2.5 Condensation and evaporation of organic vapours and sulfuric acid

Formation and evolution of either natural or anthropogenic aerosols are dependent on gas-to-particle phase transition via nucleation, condensation and evaporation (Tsang and Brock 1982; Wagner 1982) and further on coagulation (von

Smoluchowski 1918; Fuchs 1964). In ARCA, the condensational growth of particles due to organic vapours and sulfuric acid is defined by their gas and particle phase concentrations and pure liquid saturation vapour pressures $p_{sat}$ (sulfuric acid is treated as non-evaporating vapour with $p_{sat} = 0$). ARCA employs the mass conserving, Analytical Predictor of Condensation (APC) scheme (Jacobson 1997a; Jacobson 2002), describing the condensational transfer of a gas phase compound $q$ onto particles in size bin $i$, as the change in particle phase composition $c_{q,i}$, in a time interval $\Delta t$:

$$\frac{dc_{q,i,t}}{dt} = k_{q,i,t-\Delta t}\left(C_{q,t} - S'_{q,i,t-\Delta t}\,C_{q,eq,t-\Delta t}\right) \tag{4}$$





where $C_{q,t}$ is the gas phase concentration of compound $q$ at time $t$, $S'_{c,i,t-\Delta t}$ is the equilibrium saturation ratio of the condensing gas, $C_{q,eq,t-\Delta t}$ is the pure compound saturation vapour concentration over a flat surface and $k_{q,i,t-\Delta t}$ is the mass transfer rate between the gas phase and all particles of size $i$ (Jacobson 1997a). The mass transfer rate $k_{q,i,t} = n_i 2\pi d_i D_{eff,q,i}$ [s$^{-1}$], where $n_i$ is the number concentration of particles in size bin $i$, $d_i$ is the diameter of the particle in size bin $i$, and $D_{eff,q,i}$ is the diffusion coefficient of compound $q$ with particles of size in bin $i$ (Jacobson, 1997b). The equilibrium saturation ratio is calculated

using the Köhler equation, which combines both the Kelvin and solute effect. The Kelvin effect accounts for the changes in saturation vapour pressure over the particle due to surface curvature, with small particles having larger saturation vapour pressures. The solute effect, or Raoult's law, describes the change in saturation vapour pressures for an ideal solution with a mixture of compounds in the particle droplet. Overall, the equilibrium saturation ratio for a compound is obtained by multiplying the solute effect, expressed as the molar fraction of the compound and the Kelvin term:

$$S'_{q,i,t} = x_{q,i} \exp\left( \frac{4\sigma_{q,p} m_q}{R_g d_i T_t \rho_{q,p}} \right) \tag{5}$$

where $x_{q,i}$ is the organic molar fraction of compound $q$ in a particle of bin $i$, $\sigma_{q,p}$ is the pure liquid surface tension, $m_q$ is the molar mass of the compound, $R_g$ is the universal gas constant, $T_t$ is the temperature at time $t$, $\rho_{q,p}$ is the liquid density of the pure compound $q$ and $d_i$ is the diameter of particle in bin $i$.

Forward Euler method is used to integrate equation (4) over incremental time steps $\Delta t$, giving the change in particle phase concentrations $\Delta c_{q,i,t+\Delta t}$. The noniterative solution to equation (4) can be written as:

$$c_{q,i,t} = c_{c,i,t-\Delta t} + \Delta t\, k_{c,i,t-\Delta t}\left( C_{q,t} - S'_{q,i,t-\Delta t} C_{q,s,t-\Delta t} \right) \tag{6}$$

Equation (6) relies on final gas concentrations $C_{q,t}$, which are constrained by the mass balance equation

$$C_{q,t} + \sum_{i=1}^{N_{bins}} c_{q,i,t} = C_{q,t-\Delta t} + \sum_{i=1}^{N_{bins}} c_{q,i,t-\Delta t} \tag{7}$$

where $N_{bins}$ is the number of particle size bins. Combining equation (6) with the mass balance equation (7), the change in mass composition is calculated, which is then passed to the particle redistribution module. The final particle composition $c_{q,i,t}$ at each time step is constrained between max(0, $c_{q,i,t}$) to prevent evaporation from exceeding the total mass existing in each particle size bin (Jacobson 1997b).

The most crucial factor governing the condensational growth in the APC scheme is the compound specific pure liquid saturation vapour pressure $p_{sat}$. The structures of all compounds originating from MCM are available as SMILES codes, and ARCA includes a tool that can be used to extract the temperature dependent $p_{sat}$ information using the methods in the UmanSysProp (http://umansysprop.seaes.manchester.ac.uk) database. The tool converts the data to A and B parameters of the Antoine equation: $\log_{10}(P_{sat})$ = A – B/T, where $p_{sat}$ is in atm and T is the temperature [K].

There are often orders of magnitudes discrepancies (ranging from $10^{-13}$–$10^{-5}$ atm) in the estimated pure liquid saturation vapour pressures of species when using different methods (Valorso et al. 2011). Therefore, ARCA includes two different files for saturation vapour pressures; one obtained using the methods described in Nannoolal et al. (2008) and another using the EVAPORATION (Compernolle et al. 2011). The files include only those compounds whose $p_{sat} \leq 10^{-6}$ atm (at 293.15 K), as more volatile compounds have negligible contribution in particle growth. In both files, the saturation vapour pressures for

HOM's originating from PRAM are calculated using the group contribution method SIMPOL (Pankow and Asher 2008). Kurtén et al. (2016) have shown that NANNOOLAL produces low estimates of saturation vapour pressure for multifunctional compounds due to the absence of hydro-peroxide or peroxy-acid group parametrizations. SIMPOL on the other hand, has shown to be in better agreement with pure-liquid vapour pressures of multifunctional compounds calculated using COSMO-RS (Conductor-like Screening Model for Real Solvents) (Eckert and Klamt 2002; Kurtén et al. 2016). The

EVAPORATION includes a limited number of peroxides and peroxy acids (Kurtén et al. 2016), and is shown to produce the most accurate estimation of $p_{sat}$ for all compounds for which EVAPORATION is applicable (O'Meara et al., 2014). The user can always provide their own pure liquid saturation vapour pressure data (using the same file formatting: compound_name,



A, B). The GUI also contains a tool where the NANNOLAAL and EVAPORATION data can be filtered with some other threshold than $10^{-6}$ atm.


### 2.6 Coagulation of particles

Coagulation occurs when two particles collide and coalesce or form agglomerates, and results in the increase of mean particle size and decrease of number concentration in the total particle distribution, while total mass is unaffected. As ARCA is primarily intended to be used within submicron size range, the only coagulation process considered is the Brownian

(thermal) coagulation, caused by thermally induced random motion of particles. Models of Brownian coagulation have existed for over a hundred years, starting from the work by von Smoluchowski (1918). The coagulation equation (Seinfeld and Pandis 2016) is

$$\frac{\partial N_i(t)}{\partial t} = \frac{1}{2} \sum_{j=1}^{i-1} K_{j,i-j} N_j N_{i-j} - N_i \sum_{j=1}^{\infty} K_{i,j} N_j , \quad i \geq 2 \tag{8}$$

where $K_{i,j}$ is the coagulation coefficient, or coagulation kernel, between particles of size $i$ and $j$, with number concentration $N$. Coagulation coefficient $K$ depends on the size of the particles and increases with the particle size difference. Derivation of

coagulation coefficient is different in the free molecular, transition and the continuous regime, and the commonly used method to account for this is the Fuchs Form of Brownian Coagulation Coefficient (Fuchs 1964; Seinfeld and Pandis 2016) which also accounts the coagulation efficiency, or sticking coefficient, the fraction of collisions that lead to coagulation, $\alpha_c$ (in ARCA the default ALPHA_COA=1). In general coagulation coefficients contain uncertainties and can be affected by factors which are not considered in the model, such as particle shape, hardness, electric forces (whether induced or by net charge).

The thermal speed of nanosize particles is in the order of tens of meters per second, and the probability of succesful coagulation could differ substantially from 1.

Other coagulation processes, such as gravitational, turbulent or shear coagulation are not considered in ARCA. This simplification can be justified for submicron sized particles, for which they are two–three orders of magnitudes slower compared to Brownian coagulation (Seinfeld and Pandis 2016). If an estimation of the effect of these additional coagulation

pathways is needed, they could be considered as additional loss terms and handled in loss module.

### 2.7 Losses of condensable vapours and particles

ARCA considers losses of gas phase compounds and particles in separate steps, as shown in Fig. 1. The compounds considered in the vapour wall loss module are the same set of organic compounds that are condensing on the particles. The wall loss is a reversible process characterized by two reaction rates $k_{gas\to wall}$ and $k_{wall\to gas}$

$$\frac{dC_{q,g}}{dt} = -k_{gas\to wall} C_{q,g} + k_{wall\to gas} C_{q,w},$$
$$\frac{dC_{q,w}}{dt} = k_{gas\to wall} C_{q,g} - k_{wall\to gas} C_{q,w} \tag{9}$$

where $C_{q,g}$ and $C_{q,w}$ are the total concentrations of compound $q$ in gas phase and wall, respectively. ARCA uses Fortran DVODE to solve equations 9 for each compound at each time step. Rate constant $k_{gas\to wall}$ is derived from kinetic gas theory assuming a well stirred chamber and is limited either by diffusion near the wall, or uptake by the wall itself ( McMurry and Grosjean 1985)

$$k_{gas\to wall, q} = \frac{A_{ch}}{V_{ch}} \alpha_w \bar{v}_q \left( 4 + \frac{\pi \alpha_w \bar{v}_q}{2\sqrt{(k_e D_q)}} \right)^{-1} \tag{10}$$

where $A_{ch}$ and $V_{ch}$ are the area and volume of the chamber, $\alpha_w$ the accommodation coefficient, $\bar{v}_q$ and $D_q$ the average thermal

speed and diffusivity of molecule of compound $q$, respectively, and $k_e$ the eddy diffusion coefficient. $\alpha_w$ (ALPHAWALL) is a property of the chamber wall, whereas $k_e$ (EDDYK) is a description of the turbulent conditions in the chamber. In general, we



should assume that $\alpha_w$ is different for each compound, but here a constant value is used for all compounds. The rate $k_{wall\rightarrow gas}$ is derived from a steady state equilibrium where $\frac{dC_{q,g}}{dt}=\frac{dC_{q,w}}{dt}=0$ and $C_{q,g}=C_{q,sat}$, then

$$k_{wall\rightarrow gas}=k_{gas\rightarrow wall}\frac{C_{q,sat}}{C_{q,w,eqv}} \tag{11}$$

where $C_{w,eqv,q}$ is the equivalent mass concentration on the wall (Pankow 1994; Matsunaga and Ziemann 2010). It is assumed
that the activity of compound $q$ is always 1, rendering $C_{w,eqv,q}$ to a constant $C_{w,eqv}$ (`CW_EQV`). The value of $C_{w,eqv}$ is not known, but the sensitivity to the loss of low volatility vapours is small above values ~1μmol/m³ (Zhang et al. 2014). The default $C_{w,eqv} = 40$ μmol/m³, corresponding to 10 mg/m³ for an organic molecule with a mass of 250 g/mol.

Aerosol losses are considered as irreversible deposition, and the first order loss rates [1/s] can either be a constant value or read from a file as (time and) size resolved values (which will be linearly interpolated for model times and bin diameters). If
the losses are not known, they can also be approximated using parametrization from Lai and Nazaroff (2000), which considers the different deposition velocities to upwards, downwards and vertical surfaces. The necessary input is the floor area, height and the friction velocity of the chamber (`CHAMBER_FLOOR_AREA`, `CHAMBER_HEIGHT` and `USTAR`, respectively). The last is used to characterize the near-surface turbulent flow, and can be estimated from the air flow velocity in the chamber with Clauser-plot method (Bruun 1996), or treated as a fitting parameter. Whichever way is used to derive the loss rate $k_{dep,i}$,
the number of particles lost from bin $i$, $\Delta N_{i,\Delta t}$, in time step $\Delta t$ are calculated by

$$\Delta N_{i,\Delta t}=N_{i,t}\left(1-\exp\left(k_{dep,i}\Delta t\right)\right) \tag{12}$$

The model saves the mass composition of particles lost to walls (in kg/m³) as a cumulative sum. Since the vapour wall losses are reversible, the model saves the mass flux to and from the walls (in kg/m³/s, positive value indicates gas to wall flux), calculated as an average over the time interval for saving results.

**2.8 Model output**

ARCA saves most of the output from the simulations in three compressed NetCDF4 (Network Common Data Form) files. The time resolution of the output can be defined in model seconds (`FSAVE_INTERVAL`), or as number of saved instances (`FSAVE_DIVISION`). The files contain time series of the environmental variables and nucleation rates and the names of the used ACDC systems (*General.nc*), gas concentrations of the complete chemical set (*Chemistry.nc*) and aerosol number concentration, size, coagulation sink, particle growth (by condensation module) and loss rates, vapour concentrations in
particle (size resolved) and gas phase and the vapour fluxes to and from the particle phase and walls (*Particles.nc*). All files contain basic attributes of the model configuration, such as the name of the chemistry module used, user supplied description, name of the output directories and INITFILE, (real) date and time of the simulation.

When the model is run, a copy the INITFILE which was used to initialize the model is saved (*InitBackup.txt*). This file can be loaded in the GUI or used as such to repeat the simulation, provided that other input files are the same. Time series of
particle number concentrations in normalized $\Delta N/\Delta(\log_{10}(d_p))$ and linear scale, and a list of condensing vapours is provided for convenience. If the model was run from the GUI, the screen output of the numerical model is also saved (*runReport.txt*). A complete list and description of the output variables and files is shown in Table 2.

**Table 2: Description the output file contents, dimensions and units.**

| Variable name | Dimensions | Units | Description |
|---|---|---|---|
| **All NetCDF files** | | | |
| TIME_IN_SEC | time | s | Model time in seconds |
| TIME_IN_HRS | time | h | Model time in hours |
| **General.nc** | | | |


| Variable name | Dimensions | Units | Description |
|---|---|---|---|
| [Time-dependent input variables] | time | [in units] | All Time-dependent input variables *which are used in the simulation,* are saved in the output after unit conversions, and it is good practice to check that the values are as intended. |
| J_ACDC_[1–5]_CM3 | time, 4 | $s^{-1}$ $cm^{-3}$ | Four elements are formation rates of summed, neutral, positively and negatively charged clusters ($J_{sum} = J_{neut} + J_{pos} + J_{neg}$). |
| J_ACDC_SUM_CM3 | time | $s^{-1}$ $cm^{-3}$ | Summed total formation rates from all ACDC systems used |
| J_TOTAL_CM3 | time | $s^{-1}$ $cm^{-3}$ | Summed total formation rates from all particle formation methods. This is the rate which is used in the aerosol module |
| CS_CALC | time | $s^{-1}$ $cm^{-3}$ | Condensation sink of sulfuric acid as calculated from the modelled PSD |
| **Chemistry.nc** | | | |
| [COMPOUND_NAMES] | | $cm^{-3}$ | The full CH_GAS vector (CH_GAS is an internal variable containing concetrations of all chemical compounds in the chemistry |
| [REACTIVITIES] (if calculated) | | $s^{-1}$ | The effective reactivities (inverse lifetimes) of selected compounds, if defined in the chemistry |
| **Particles.nc** | | | |
| VAPOURS | composition | | Names of the compounds that can go to particle phase. They are also listed in the text file CondensingVapours.txt. |
| NUMBER_CONCENTRATION | time, bins | $cm^{-3}$ | Particle number concentration. They are NOT normalized. Particle number concentration is also saved in two text files: Particle_conc.dat, which is exact copy of this variable, and Particle_conc.sum, which ARE normalised by $\Delta N/\Delta(\log_{10}(d_p))$. The text files contain time stamps in the first column and diameters in the first row. Additionally, the .sum-file contains the total particle concentration in the second column. |
| INPUT_CONCENTRATION | time, bins | $cm^{-3}$ | Particle number concentration of the input PSD (including the (multi)modal PSD), after it has been converted to model diameter grid. They are NOT normalized. |
| DIAMETER | time, bins | m | Particle nominal diameters |
| GROWTH_RATE | time, bins | nm/h | Instantaneous condensational growth rate of the particles. Positive = particles are growing. Note that this is not the same (even if it often is close) as what is obtained by appearance time or mode-fitting methods, when growth rate is calculated from observations. |
| COAG_SINK | time, bins | $s^{-1}$ | Instantaneous sink of particles due to coagulation with larger particles |
| MASS | time, bins | kg | Particle mass |
| PARTICLE_COMPOSITION | time, bins, composition | kg | Singe particle mass, based on the volume. For the MA PSD method, this uses the actual particle diameter, not the nominal diameter. |
| MASS_FLUX_ON_PAR | time, composition | kg $m^{-3}$ $s^{-1}$ | Flux of vapours to the particles, averaged over `FSAVE_INTERVAL`. Positive = flux from gas phase to walls. Saved only if vapour wall loss module is used. |
| DEPOSITED_PAR_COMP | time, composition | kg $m^{-3}$ | Cumulative sum of composition lost to walls. Saved only if particle wall loss module is used. |
| PARTICLE_LOSS_RATE | time, bins | $s^{-1}$ | Size-resolved instantaneous loss rate of particles. Saved only if particle wall loss module is used. |
| MASS_FLUX_ON_WALLS | time, composition | kg | Flux of vapours to the chamber walls, averaged over `FSAVE_INTERVAL`. Positive = flux from gas phase to walls. Saved only if vapour wall loss module is used. |
| **Other output files** | | | |
| InitBackup.txt | - | - | Contains a backup of the INITFILE, and can be used to repeat the simulation |
| NMLS.conf | - | - | Dump of all namelist variables, even if they were not included in the INITFILE |
| Particle_conc.sum and .dat | time, bins | $cm^{-3}$ | See NUMBER_CONCENTRATION |
| runReport.txt | - | - | The screen ouput of the numerical model. Only created when run from the GUI. From terminal *tee* command can be piped with the program call. The Create batch tool writes a bash script which has the *tee* command properly configured. |
| CondensingVapours.txt | - | - | See CONDENSABLES |
| optimization.txt | - | - | Output of the time step optimisation. Records the times and processes that were causing the time step increase or decrease. |


## 3 Using the model – graphical user interface (GUI)

ARCA box has an extensive online user manual, along with tutorials of installation and example cases. The manual is written completely from perspective of the user interface, and the underlying scientific base of any procedure is mentioned only if relevant to the instruction. Therefore, we will here introduce the GUI only broadly but emphasize once more that the

GUI is one of the main features setting ARCA apart from other similar models.



Typically, a model such as ARCA is used for sensitivity studies and is run multiple times, changing various parameters. It is very easy to lose track of the differences in different simulations, even when the simulation settings are not hardcoded. A user interface is a valuable aid in organizing the model options in a way that helps the user to have a good visual control of the workflow. Although a good user interface does not directly equal good science, we hope that it leads to better science.

ARCA's GUI enables this by showing the numerous simulation settings in the relevant context, thus making it easier to check that the input (e.g., units, files, processes in use etc.) are correct, by automatizing file and directory naming and creation, by increasing reproducibility with logging and automatic backup of executed simulation settings, by making inspection of the results consistent with the plotting tools tailored for model output and by providing guidance with direct links to the corresponding page in the online manual, just to name a few examples. While the numerical model is often run

directly from the GUI, with a valid INITFILE the compiled Fortran program can always be run from the command line, without the GUI. In fact, this is the best way to use the model in batches of simulations, such as would often be done on a remotely run High Performance Computer.

### 3.1 Selected features of the GUI

To give an overview of the GUI, figures 5–7 show selected screenshots of the GUI window. Figure 5 shows the tab *General*

*options* (left panel), which contains options for naming of output, paths to input and output files, model time step, tool for creating batches of simulations and toggles for the main modules. The right panel shows the options related to aerosol module, such as size range and resolution of the PSD grid, its initialization with either measurements from file or using (multi)modal lognormal distribution, and duration and size range of initialization. Figure 6 shows the input options for the time dependent variables, and Fig. 7 shows examples of the output plotting tools such as surface plots and particle mass and

size distribution plots. A useful tool is the *Live update* option, which shows the evolving particle size distribution surface plot during the simulation.

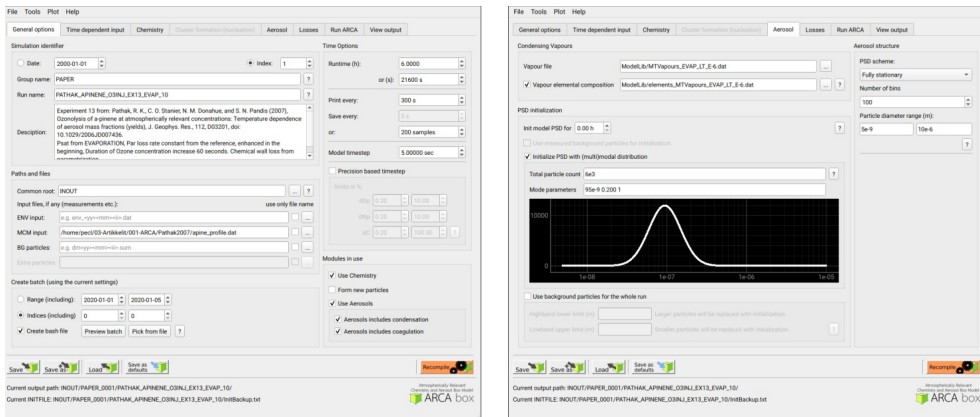

**Figure 5: Screenshots of the GUI showing the tabs *General input* (left) and *Aerosol* (right), where the black plot is the user defined parametric initial particle size distribution that is distributed to the selected size bins. The settings shown here are from the simulation shown in Fig. 10.**



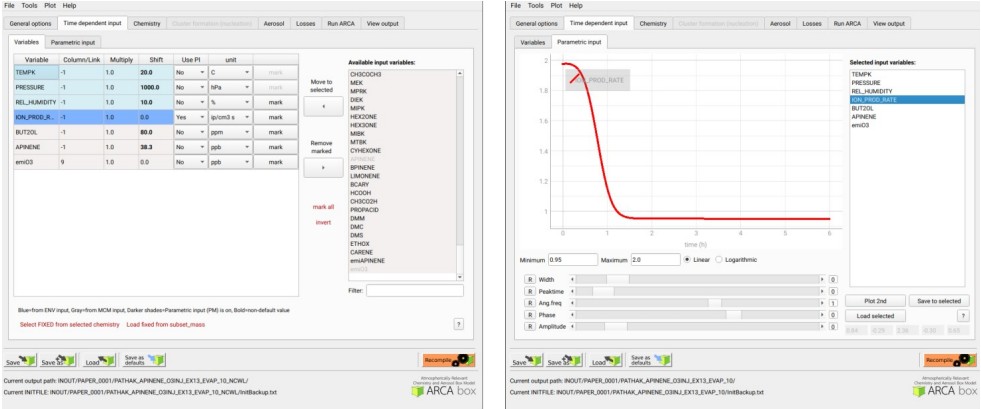

**Figure 6: Example of the time dependent input definition: Selecting the necessary model input, their sources and units (left); parametric input tool (right).**

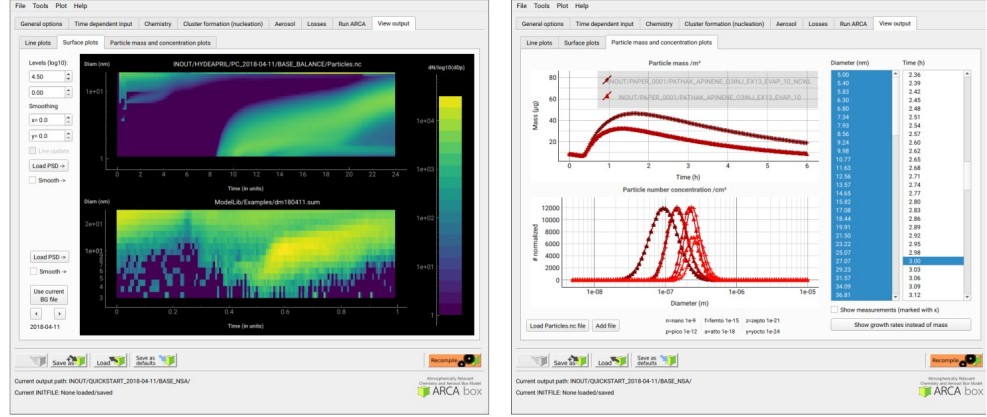

**Figure 7: Examples of the plotting tools of the model output: surface plots of the modelled (left, top plot) and initialization PSD (left, lower plot); On the right comparison of two simulations, with total particle mass (right, top plot) and size distribution at three time instances (right, lower plot).**

## 4 Verification and evaluation of main modules

The aim of this section is to verify that the modules in ARCA box perform programmatically as intended. We must stress that the validity of the results of any model is strongly dependent on the input parameters used. Here the reader is reminded of the most crucial parameters in addition to the user supplied concentrations:

– for the chemistry module the chemical reaction sets (and accompanying kinetic rate coefficients), spectral data

– for the formation rate module (ACDC) the Gibbs free energies

– for condensation and vapour loss modules the pure liquid saturation vapour pressures

These are provided in the default ARCA installation as a working example, and while they are from published sources, they are not intended to be used in all conditions and locations. Instead, the user should use the tools in the GUI and online manual to collect and prepare their own set of input parameters and submodules, and justify their use based on their simulation conditions. The manual has detailed instructions on how to acquire and format the input data, construct a chemistry module and update the ACDC systems. The tests shown here are simplified cases whose purpose is to show that the calculations in the model are done correctly. The last test is a comparison against a chamber experiment, which utilises



nearly all the modules and therefore connects all the individual processes. The settings used in this simulation is shown in Appendix C and serve as an example of an INITFILE.

## 4.1 ACDC


For any given cluster system, the formation rates in ACDC depend largely on the evaporation rates of the clusters, generally calculated from the input Gibbs free energies. Figure 8 shows formation rates calculated with the four ACDC simulation systems, two for $H_2SO_4$–$NH_3$ and two for $H_2SO_4$–DMA. The systems calculated with the RICC2 (RICC2/aug-cc-pV(T+d)Z//B3LYP/CBSB7) method are described in Olenius et al. (2013) and can be found from the ACDC repository

(github.com/tolenius/ACDC). The DLPNO level $NH_3$ system (DLPNO-CCSD(T)/aug-cc-pVTZ//ωB97X-D/6-31++G**) was used in Besel et al. (2020), whereas the DLPNO DMA system was used in Myllys et al. (2019). Figure 8a shows the significance of ion-mediated clustering in the $H_2SO_4$–$NH_3$ system, as is apparent when comparing the total formation rates with and without the presence of ions. It also underlines the already mentioned notion that the charge of the outgrowing clusters does not necessarily correspond their pathway inside the system. The $H_2SO_4$–DMA cluster formation (Fig. 8b)

shows weaker sensitivity to the presence of ions.

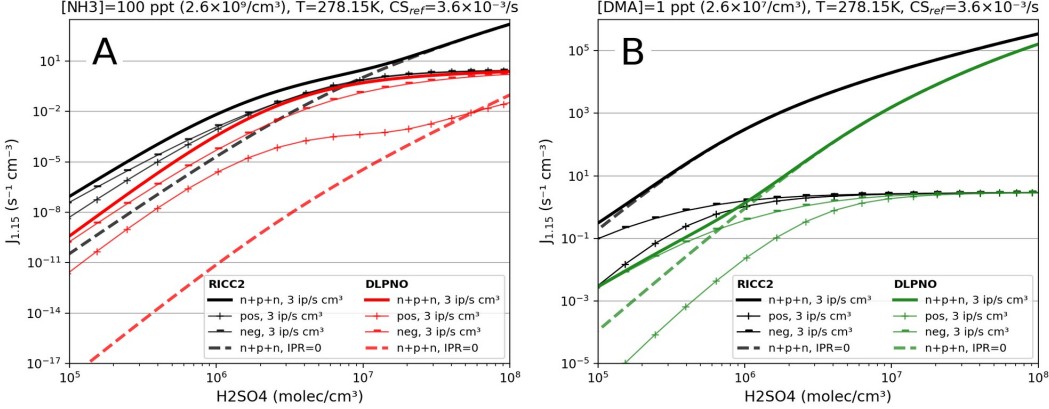

**Figure 8: Steady-state particle formation rates from the two $H_2SO_4$–$NH_3$ (panel A) and two $H_2SO_4$–DMA systems (panel B). Both chemistries include data from RICC2 and DLPNO level of theories. Thick lines show the total formation rate, solid lines are in the presence of ions, dashed lines without ions present. All simulation used the same temperature (5° C) and external cluster losses (3.6×10⁻³/s).**

## 4.3 Verification of the coagulation module

The calculations of the Brownian coagulation module were verified in two parts. Figure 9a shows the size dependant

coagulation coefficients calculated by ARCA (for comparison, see figure 13.5 in Seinfeld and Pandis (2016)) whereas Fig. 9b compares the evolution of a particle size distribution with an analytical solution to the coagulation of a polydisperse particle volume distribution. The solution applies constant, size-independent coagulation coefficient $10^{-9}$/cm³/s (defined by the initial concentration $N_0 = 10^6$/cm³ and characteristic time $\tau_c = 2000$ s), and for this test ARCA's calculated coefficients were replaced with the same constant coagulation coefficient (for comparison, see figure 13.6 in Seinfeld and Pandis

495 (2016)).





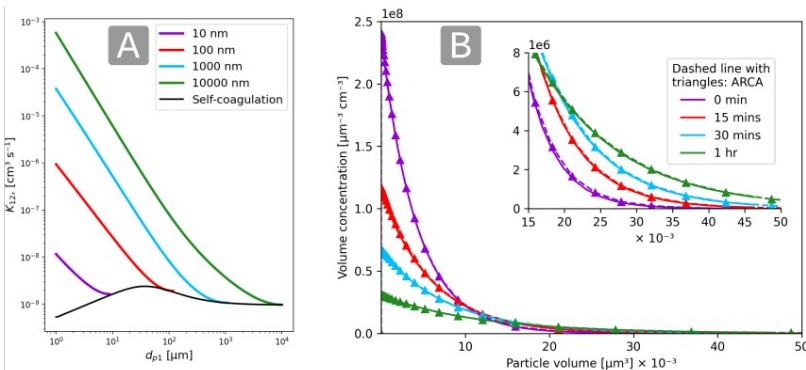

**Figure 9: Coagulation coefficients used in ARCA box for four different particle diameters (panel A), with sticking coefficient $\alpha_g$=1. Comparison with analytical coagulation solution (panel B).**

### 4.2 Chemistry, condensation and loss routines

We simulated α-pinene oxidation in high ozone and low OH chamber environment, similar as in experiment 13 in Pathak et al. (2007). The 10 m³ Teflon bag chamber was initially filled with seed particles, α-pinene and 2-butanol as OH scavenger.

Then $O_3$ was introduced to the chamber. The reported mean particle loss rate (0.3/h) was expanded to size and time dependant profiles using Pierce et al. (2008), who measured the losses in the same chamber in more detail. Figure 10 shows the resulting time series with moderate wall loss of condensing vapours (ALPHAWALL = 5×10⁻⁵, CW_EQV = 40 μm/m³ and EDDYK = 0.05 s⁻¹). The final SOA mass yield of 0.168 (final SOA mass / initial α-pinene mass) calculated by ARCA agrees well with the reported 0.17. Without evaporation from the particles, invoked by the accumulation of vapours to walls and the

consequential supersaturation decrease, the calculated yield is 0.177. In the chemistry scheme built for this test, ozone concentration was regulated by emission rate (similarly as would be in a chamber experiment), and a mixing time of 1 minute was achieved by tuning the emission rate (in the end 2 ppb/s) so that the $O_3$ concentration increases from 0 to the target (250 ppb) in about 200 seconds, and then turned on only to keep the concentration constant. Without chemical wall loss the simulated yield is 0.286, overshooting the reported yield by a factor of 1.7. The pure liquid saturation vapour

pressures used were derived using the EVAPORATION method. The chemistry was acquired from MCM and amended with the PRAM, and $p_{sat}$ data was downloaded from UmanSysProp, using the tools in ARCA box.



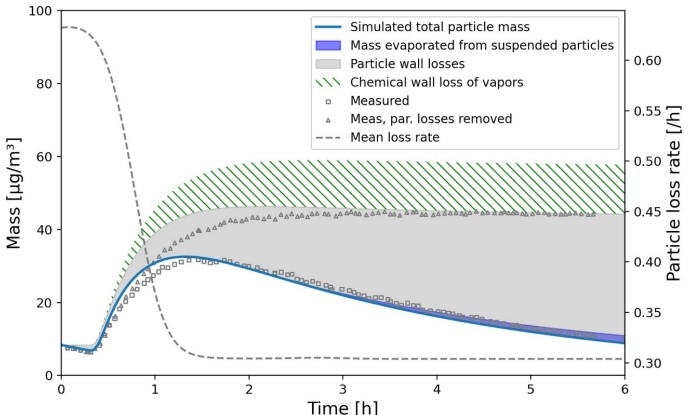

**Figure 10: Comparison against α-pinene oxidation experiment (exp. 13, Pathak et al. 2007). Size-resolved aerosol wall losses were measured for the same chamber in Pierce et al. (2008) Moderate chemical wall loss is calculated with parametrisation from ARCA box using `ALPHAWALL = 5×10⁻⁵`, `CW_EQV = 40 μm/m³` and `EDDYK = 0.05 s⁻¹`.**

## 5 Technical information of ARCA box

### 5.1 Licensing

ARCA box is licensed under GPL 3.0, and the licensing statements (also those from auxiliary software) are included in the source code. The source code includes software that is not written by authors of ARCA, namely the Fortran VODE solver ("In the Public Domain", or IPD), ACDC (GPL 3) and KPP, which is not strictly necessary for ARCA but provided for convenience (GPL 3), as the original KPP is not able to compile very large chemistry schemes or long variable names.

### 5.2 System requirements

ARCA's numerical model is written in Fortran, and the user interface in Python 3. These environments must be installed and properly working on the computer. Also, since the output data is mainly saved in NetCDF 4 files, this software – along with its Fortran and Python libraries must be installed prior to compilation and use. ARCA box has been developed on Linux platform, but due to the cross-platform nature of Python and the availability of Gnu Fortran for all three major OS's, the model has successfully been installed and used on all of them (on Windows GFortran is used through Cygwin and on

MacOS through Xcode). The model has not been tested with Intel Fortran but as it is usually compatible with GFortran, we expect no major issues. The ARCA online manual has step-by-step instructions and videos for the installation of the prerequisite environments as well as the model itself. It also contains solutions to installation problems which have been reported to us. After the necessary environments are working, installation and compiling the model itself is straightforward with the included Python installer script.

### 5.3 Code availability

The ARCA model source code, described in this paper, is publicly accessible as a frozen archive (available at https://doi.org/10.5281/zenodo.6358578). However, the most recent, constantly updated version is available for download upon request in www2.helsinki.fi/en/researchgroups/multi-scale-modelling/arca. The users are asked to provide their email address and a very brief overview of the intended field of study with ARCA. This information is used to inform of any future





updates, fixes and other news regarding the model, as well as give the ARCA model development group information of the different uses of the model. After the registration the user will be submitted a Git pull token to the private Gitlab repository. This token can be used later at any point for updating or reinstalling the code.

ARCA's user manual is in Wiki format, found in https://wiki.helsinki.fi/display/arca. There are also links from the GUI directly to the relevant parts in the online manual. In addition to the manual, there are tutorial videos, and troubleshooting

instructions. The manual is updated continuously as the model is further developed, but older states for previous versions are saved in pdf format, available for download at the site.

## 6 Future developments

In its current state, we see ARCA as a robust base, a platform that packages established theories and knowledge of the central processes of this domain in a user-friendly and extendable program. Still, many known processes are for now omitted

from the model, and the model will be developed further. This is aided by the fact that ARCA is one of the primary zero-dimensional process models used by the authors. Current work with model development is concentrating on implementing an inorganic thermodynamic module, similarly as in ADCHEM and ADCHAM (Roldin et al. 2011; Roldin et al. 2014). This will enable calculations of size-resolved aerosol hygroscopic growth, acidity (pH) and the saturation concentrations of inorganic acids such as $HNO_3$, HCl and MSA (Methanesulfonic acid, $CH_3SO_3H$). This information will then be transferred to

the condensation/evaporation module which will use the analytical predictor of dissolution (APD) method (Jacobson 1997b) to solve the gas-particle partitioning. Another area of improvement is the addition of charged particles, which is a significant factor in chamber wall losses. These extensions will be available after evaluation in the next version of ARCA. To compliment the current particle size distribution methods (FS and MA), we also plan to add a hybrid PSD representation (Chen and Lamb 1994; Pichelstorfer and Hofmann 2015). It consists of a fixed size bin grid where the concentrations are

described by uniform distributions whose width can vary within each bin. Thus, upon growth or shrinkage, only a fraction of the population is moved to the neighbouring grid cell. This prevents numerical diffusion and avoids "pits" and "peaks" in the PSD output.

ARCA box has already been used and tested by several groups, and the feedback has helped further develop the model and its documentation. The approachable interface and model structure has been a great asset – on one hand it has helped to gain

new users, and on the other hand it has helped us to improve the usability and stability, resulting to updates for the whole user community. We are looking forward to the future, where the users of ARCA further participate in the model development by sharing their experience, needs, ideas and even code additions.

## Author contribution

PC and CX were the principal programmers and jointly coordinated the writing of the numerical model. PC wrote the user

manual and the graphical user interface. LP wrote the PSD module and the time step optimization. PZ provided the chemistry modules and the supporting chemistry tools for the user interface. TO provided the ACDC plug-in. PR provided the loss routines. MB wrote the chemistry interface. All authors contributed to this manuscript by way of writing and commenting.

## Acknowledgements

PC and CX share first authorship of this paper. We acknowledge the ACCC Flagship, funded by the Academy of Finland (grant number: 337549) and the computational resources from CSC – IT Center for Science, Finland. PC gratefully acknowledges the University of Edinburgh and University of Helsinki Partnership Programme on Forests. TO gratefully

acknowledges the Swedish Research Council VR (grant no. 2019-04853) and the Swedish Research Council for Sustainable Development FORMAS (grant no. 2019-01433) for financial support. PR gratefully acknowledges the Swedish Research
Council VR (grant no. 2019-05006), the Swedish Research Council for Sustainable Development FORMAS (grant no. 2018-01745), the Crafoord foundation (grant no. 20210969) and the strategic research area MERGE for financial support.

**Competing interests**

The authors declare that they have no conflict of interest.

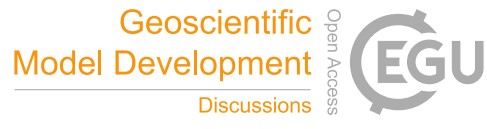



## Appendix A: List of variables used in the INITFILE

The user definable model options are briefly described here. **All options listed here can be defined in the graphical user interface (GUI)**. We want to emphasize that *there is no need to configure ARCA by manually editing the INITFILE.* In fact, this would probably lead to unintended outcomes, as some sanity checking of the options is done in the GUI. Additionally, the GUI contains tools, tooltips, help links and visualisation of the options. If necessary, for example for model development, the user can also insert any text input to the INITFILE from the GUI, and there is no need to part from the GUI workflow

even when an option is not (currently) available in the GUI. All settings, even the raw input is always saved in the INITFILE written by the GUI, and will be available when an INITFILE is loaded in the GUI.

**Table A1: Model variables which can be set from the GUI and INITFILE**

| Unit | Variable type | Variable name | Description, input options. **Bold** font shows the recommended value (if applicable) |
|------|---------------|---------------|---------------------------------------------------------------------------------------|
| **NML_TIME** | | | |
| h | REAL | RUNTIME | Duration of the simulation. The GUI accepts seconds and converts them to hours before writing the INITFILE. The simulation is always started from time 0, midnight, but the clock can be moved forward with NML_CUSTOM->START_TIME_S |
| s | REAL | DT | Integration time step. When USE_SPEED is TRUE, it is the minimum time step. |
| s | REAL | FSAVE_INTERVAL | Time interval (simulation time) for the output file writing |
| s | REAL | PRINT_INTERVAL | Time interval (simulation time) for the screen output |
| | INTEGER | FSAVE_DIVISION | when > **0**, the output files will contain approximately FSAVE_DIVISION timestamps |
| | CHARACTER | DATE | 'yyyy-mm-dd', alternative to INDEX, used for naming directories and calculating solar angle. |
| | CHARACTER | INDEX | 'xxxx' four digit index, alternative to INDEX, used when sun angle is not relevant. |
| **NML_FLAG** | | | |
| | LOGICAL | CHEMISTRY_FLAG | **T**/F whether chemistry module is used or not |
| | LOGICAL | CHEM_DEPOSITION | **T**/F whether vapour wall losses are considered |
| | LOGICAL | ACDC | **T**/F whether nucleation module (ACDC) is used or not. Does not affect parametric nucleation or formation rate that is sent in as time-dependent variable |
| | LOGICAL | ACDC_SOLVE_SS | T/**F** whether ACDC is solved to steady state |
| | LOGICAL | AEROSOL_FLAG | **T**/F whether aerosol module is used or not (nucleation module is unaffected) |
| | LOGICAL | CONDENSATION | **T**/F whether gas-particle partitioning is considered |
| | LOGICAL | COAGULATION | **T**/F whether aerosol coagulation is considered |
| | LOGICAL | DEPOSITION | **T**/F whether particle losses are considered |
| | LOGICAL | MODEL_H2SO4 | **T**/F a convenience option, if T, H2SO4 concentrations are calculated by chemistry, if F, input values are used in all modules. |
| | LOGICAL | PRINT_ACDC | T/F if T, ACDC modules (in use) print on screen the cluster concentrations inside the system at each PRINT_INTERVAL |
| | LOGICAL | OPTIMIZE_DT | T/F whether time step optimization is used |
| | LOGICAL | ORG_NUCL | T/**F** whether parametrisation for organic nucleation is used |
| | LOGICAL | AFTER_CHEM_ON | T/**F** whether subroutine AFTER_CHEM in custom_functions.f90 is called after chemistry step. AFTER_CHEM is a dedicated injection point for customized code. |
| | LOGICAL | AFTER_NUCL_ON | T/**F** whether subroutine AFTER_NUCL in custom_functions.f90 is called after nucleation step. AFTER_NUCL is a dedicated injection point for customized code and is executed before total formation rate is applied in the aerosol module. |
| | CHARACTER | FILE_TIME_UNIT | **'day'**,'hrs','min','sec'. The time unit used in the input files for environmental, inorganic and organic variables. |
| | CHARACTER | LOSSFILE_TIME_UNIT | **'day'**,'hrs','min','sec'. The time unit used in the input files for the particle loss rate file. |
| **NML_PATH** | | | |
| | CHARACTER | INOUT_DIR | The root directory (relative to the executable, or absolute path) where data is saved. This must exist before starting the model |




| | | | |
|---|---|---|---|
| | CHARACTER | CASE_NAME | The directory which contains the runs (designated with RUN_NAME). These will be created by the model, and the DATE or INDEX will be appended to the path name. The GUI will always show the formatted output paths and provides quick access to the directory. |
| | CHARACTER | RUN_NAME | The name of the run directory, contained in CASE_NAME. When repeating similar simulations, where some variable(s) are changed, it is enough to change the RUN_NAME to create unique output data. |

**NML_PRECISION**

| | | | |
|---|---|---|---|
| % | REAL | DDIAM_RANGE | **0.5,3.0** two-element, comma-separated list for the optimized time step tolerances considering the change in the particle diameter d$d_p$ |
| % | REAL | DPNUM_RANGE | **0.5,3.0** two-element, comma-separated list for the optimized time step tolerances considering the change in the particle number concentration d$N_p$ |
| % | REAL | DVAPO_RANGE | **0.5,3.0** two-element, comma-separated list for the optimized time step tolerances considering the change in the vapour concentration d$C$ |

**NML_VAP**

| | | | |
|---|---|---|---|
| | LOGICAL | USE_ATOMS | **T**/F whether elemental composition is used to calculate the diffusion diameter. If T, VAP_ATOMS must be provided. |
| | CHARACTER | VAP_NAMES | Path to file containing the pure liquid saturation vapour pressure data. The definition of which compounds condense on particles is based on the listed compounds in this file. Only the compounds that are found from the chemistry will be picked, so it is safe to have a larger set of compounds than actually exist in the chemistry. The compounds should be listed each on their own row, named exactly as in the chemistry, followed by molar mass, A and B term from the Antoine equation, as a *space separated* list. The GUI has a tool 'Create vapour file for aerosol module' which can extract the infdormation from UmansysProp with user supplied SMILES data. Example files are included in the default installation. |
| | CHARACTER | VAP_ATOMS | Path to file containing elemental composition of the organic compounds. This will be automatically created by the tool 'Create vapour file for aerosol module'. The file formatting follows the VAP_NAMES: a *space separated* list of compound name, molar mass, C, O, N, H, S, Cl, Br (where the chemical symbol is the number of each atom in the molecule). |

**NML_PARTICLE**

| | | | |
|---|---|---|---|
| | INTEGER | PSD_MODE | The method of PSD representation. **1** = Fully stationary (FS), 2 = Moving average, fixed grid (MA) |
| | INTEGER | N_BINS_PAR | Number of elements in the particle diameter grid |
| m | REAL | MIN_PARTICLE_DIAM | Minimum particle size |
| m | REAL | MAX_PARTICLE_DIAM | Maximum particle size. Should be large enough so that concentrations stay minimal as the boundary conditions are not constrained. |
| | CHARACTER | DMPS_FILE | Path to file containing the optional particle size distribution measurements. The time resolution is assumed to be 10 minutes, but can be changed in NML_CUSTOM→DMPS_TRES_MIN |
| h | REAL | DMPS_READ_IN_TIME | Time for initialization of the particle size distribution. The modelled particles are overwritten only in the times when there exists values in DMPS_FILE. The time resolution is assumed to be 10 minutes, but can be changed in NML_CUSTOM→DMPS_TRES_MIN. If (multi)modal PSD is used for initialization MMODAL_INPUT_INUSE = 1, the model PSD is replace every time step. |
| m | REAL | DMPS_HIGHBAND_LOWER_LIMIT | If USE_DMPS_PARTIAL is T, particles above this size continue being initialized even after DMPS_READ_IN_TIME. |
| m | REAL | DMPS_LOWBAND_UPPER_LIMIT | If USE_DMPS_PARTIAL is T, particles below this size continue being initialized even after DMPS_READ_IN_TIME. |
| | LOGICAL | USE_DMPS | **T**/F PSD will be initialized from DMPS_FILE, not from the Multimodal distribution. |
| | LOGICAL | USE_DMPS_PARTIAL | T/**F** If T, keeps overwriting the PSD at each DMPS_TRES_MIN, based on the sizes in DMPS_HIGHBAND_LOWER_LIMIT and DMPS_LOWBAND_UPPER_LIMIT. This feature is mostly used when accumulation mode particles are affected by transportation, which cannot be modelled in ARCA. For example, using a DMPS_HIGHBAND_LOWER_LIMIT of 2e-8, nucleation and early growth of particles below 20 nm can be simulated and still take into account the changes in the condensation and coagulation sink of the changing accumulation and Aitken mode. Can be terminated before the simulation with END_DMPS_PARTIAL |
| | CHARACTER | MMODAL_INPUT | '3e-8 0.15 0.5  1e-7 0.25 0.3' (shown here only as an example; the correct values depend completely on the user) Space-separated list containing *n* modal parameters. Three parameters per mode are needed, and the the resulting PSD will be formed from *n*/3 modes. The parameters are geometric mean diameter GMD [m], standard deviation of the (Gaussian) distribution and the weighing factor, used to scale the mode against other modes. It is strongly recommended that the modes are build in the GUI, which shows real-time visualization of the complete PSD, along with the total particle mass and area. |



| Unit | Type | Variable | Description |
|---|---|---|---|
| | REAL | N_MODAL | The total particle number concentration of the (multi)modal distribution [cm⁻³] |
| | INTEGER | MMODAL_INPUT_INUSE | -1/1 if -1, (multi)modal PSD is not used for initialization |

**NML_ENV**

| Unit | Type | Variable | Description |
|---|---|---|---|
| | CHARACTER | ENV_FILE | Path to file containing the (optional) time-dependent input data of the environmental variables and inorganic gases. Can be same file as `MCM_FILE`. This file is a two-dimensional, space-separated text file, where first line can (and is strongly recommended to) be a header, starting with #. Column 1 is time, and the following columns contain the values for the variables. Each line corresponds to a timestamp, and the times need not be with equal intervals. The time unit is by default day, but can be changed with `FILE_TIME_UNIT`. The column numbers link the data to the variable, and this is best done in the GUI. The units of the values must correspond to the ones shown in `MODS(i)%UNIT`, see below in NML_MODS, and are defined in the GUI. Same column can be linked to multiple variables, as they can be additionally modified using the `MODS(i)%SHIFT` and `MODS(i)%MULTI`. The GUI has a tool to print the header of the file to help in assigning the data to the model. If the time series of a variable is linked to another variable, this can be done in the GUI by using the variable name instead of the column number. The unit of the linked variable will be defined by the source variable, but the linked variable can be further modified with `MODS(i)%SHIFT` and `MODS(i)%MULTI`. |
| [s⁻¹] | CHARACTER | LOSSES_FILE | Path to file containing the (optionally time and) size resolved aerosol loss rates, using similar formatting  The size and time space will be *linearly* interpolated. A constant time and size independent loss rate can be given by writing the value with prefix '#', for example, #0.003 will result in 0.003/s loss rate. |
| m² | REAL | CHAMBER_FLOOR_AREA | Chamber floor area, used in the wall loss parametrizations. |
| m | REAL | CHAMBER_HEIGHT | Chamber height, used in the wall loss parametrizations. |
| 1/s | REAL | EDDYK | **0.05** Coefficient of eddy diffusion, describes turbulence in the chamber |
| m/s | REAL | USTAR | **0.05** Chamber friction velocity, affects particle wall losses |
| - | REAL | ALPHAWALL | **10e-5** Wall loss accommodation coefficient, wall/component property, assumed constant |
| mol/m³ | REAL | CW_EQV | **40e-6** Equivalent mass concentration of the wall, equilibrium wall vapour concentration |
| [], or W m⁻² nm⁻¹ | CHARACTER | SPECTRUMFILE | Path to the file containing spectral data. The GUI accepts same wildcards in naming as `ENV_FILE`. |
| | LOGICAL | SWR_IS_ACTINICFLUX | T/**F** If T, actinic flux (AF) function is omitted and spectral irradiance is treated as AF. The integral of the product of spectral function and `SW_RADIATION` must produce AF in W/m² |
| nm | REAL | SWR_IN_LOWER | These define the range (band) of the pyranometer used for short wave irradiance |
| nm | REAL | SWR_IN_UPPER | measurement, and is only needed if the default sea level spectrum is used. |

**NML_MCM**

| Unit | Type | Variable | Description |
|---|---|---|---|
| | CHARACTER | MCM_FILE | Path to file containing the (optional) time-dependent input data of the organic precursor molecules. Can be same file as `ENV_FILE`. For formatting information, see `ENV_FILE`. |

**NML_ACDC**

| Unit | Type | Variable | Description |
|---|---|---|---|
| | CHARACTER | ACDC_SYSTEMS | 1,1,0,0,0 comma-separated vector, 1=system in use,0=not in use |
| | CHARACTER | ACDC_LINKS(1) | 'A H2SO4 N NH3' string linking the ACDC monomers to ARCA gas names. |
| | CHARACTER | ACDC_LINKS(2) | 'A H2SO4 D DMA' string linking the ACDC monomers to ARCA gas names. |
| | ... | | |

**NML_MISC**

| Unit | Type | Variable | Description |
|---|---|---|---|
| ° | REAL | LAT | Latitude (decimal degrees) of the location for field simulation, used to calculate solar angle.  Values > 0 are N. |
| ° | REAL | LON | Longitude (decimal degrees) of the location for field simulation, used to calculate solar angle. Values > 0 are W. |
| | CHARACTER | DESCRIPTION | Maximum 1000 characters long description of the simulation. Should not include special characters. |
| | REAL | CH_ALBEDO | Ground albedo, used for calculating the actinic flux. |
| m | CHARACTER | GR_SIZES | '3e-9,10e-9,20e-9' String defining the diameter ranges used to calculate the instantaneous condensational growth rate, averaged over the size ranges. Smallest particle size is always added in the calculation. Only used in the screen output, as the output files calculate size resolved growth rates. The example would produce following screen output at each `PRINT_INTERVAL` (the values are examples): <br> `\| Sizes: 1.1 --> 3.0 --> 10.0 --> 20.0 [nm]` <br> `\| GR: 1.56E-04 1.26E-04 1.50E-04 [nm/h]` |





**NML_CUSTOM**

| | | | |
|---|---|---|---|
| | LOGICAL | USE_RAOULT | **T**/F when T, use Raoult's law, the solute effect to the saturation concentration. Should generally always be T. |
| min | REAL | DMPS_TRES_MIN | **10** Time resolution for the background particles file. |
| sec | REAL | START_TIME_S | **0** If > 0, the simulation clock will be set to this time when the integration loop starts. By default the simulation clock starts at oo:oo (midnight). A caveat is that if background aerosols are initialized for example for 1 hour, and start_time_s>3600, there will be no initialization for particles. If the initialization is set to 2 hours, particles are constrained one hour. The model assumes that the input files still start at time zero (specifically, the first time stamp is subtracted from the time vector). Therefore, if the input data must contain some values at midnight, it is sufficient to have a row of zeros (one in each column) in the beginning of the input file (after the header). |
| | REAL | DMPS_MULTI | **1e6** The conversion factor to convert the number concentration values in the background particle file from particles/cm³ to particles/m³. |
| | CHARACTER | INITIALIZE_WITH | Path to a similar simulation which is used as initialization for the current run. Works only with constant time step. |
| | INTEGER | INITIALIZE_FROM | **0** Index to the place in the files used in the initialization. If 0, last value is used. |
| | REAL | VP_MULTI | **1.0** Factor for pure liquid saturation vapour pressures. Used for sensitivity test. |
| | INTEGER | LIMIT_VAPOURS | **0** if >0, only the first *n* compounds in the VAP_NAMES are taken to condense on particles |
| h | REAL | END_DMPS_PARTIAL | 0 If >0, will terminate USE_DMPS_PARTIAL after this time (in simulation hours). |
| | LOGICAL | NO2_IS_NOX | T/**F** If T, $NO_2$ input is considered as $NO_X$. $NO_2$ concentration will be calculated inside the model by subtracting NO from $NO_X$. |
| | LOGICAL | NO_NEGATIVE_CONCENTRATIONS | **T**/F If T, all concentrations and environmental variables will be max(0,<input>) (T after it has been converted from °C to K) |
| h | REAL | FLOAT_CHEMISTRY_AFTER_HRS | Stops updating gas phase concentrations with input values after this time. Affects both concentrations and emissions |
| h | REAL | FLOAT_CONC_AFTER_HRS | Same as previous, but only for concentrations |
| h | REAL | FLOAT_EMIS_AFTER_HRS | Same as previous, but only for emissions |
| | LOGICAL | USE_RH_CORRECTION | T/F Use RH correction for $H_2SO_4$ condensation |
| s | REAL | DT_UPPER_LIMIT | **150.0,150.0,150.0** Three-element vector for the upper limits for the time steps CCH, COA and DEP, used with time step optimization. |
| | LOGICAL | ENABLE_END_FROM_OUTSIDE | **T**/F This option enables graceful termination of the simulation during the simulation. It is done by creating a file called ENDNOW.INIT in the output folder. The file must contain only one word STOP. If T, the existence of this file is checked each time the model is in PRINT_INTERVAL. If found, the output files are finalized and the simulation stops. The procedure is a one-button operation in the GUI. |
| molec/cm³ | REAL | MIN_CONCTOT_CC_FOR_DVAP | **1000** If time step optimization is used, the changes to control the time steps are not calculated for gases whose concentration is below this limit. |
| | REAL | ALPHA_COA | **1.0** sticking coefficient for coagulation |
| | LOGICAL | KELVIN_TAYLOR | T/**F** approximate Kelvin equation with first 2 terms of the Taylor series. Only used for comparison with some older models. |
| N/m², J/m³ | REAL | SURFACE_TENSION | **0.05** Common surface tension for liquid phase organic compounds |
| | CHARACTER | HARD_CORE | '**GENERIC**' Name of the non-evaporating generic composition, used for initialized particles and those from nucleation. Can be thought as primary particles. Must be the last item in the VAP_NAMES file. |
| kg/m³ | REAL | ORGANIC_DENSITY | **1400** Common liquid phase density for organic compounds |
| kg/m³ | REAL | HARD_CORE_DENSITY | **1400** Density of the GENERIC |
| | REAL | NPF_DIST | **1.15** multiplied with MIN_PARTICLE_DIAM to get the upper diameter where the nucleated particles are distributed. Majority of the clusters will be assigned in MIN_PARTICLE_DIAM. |

**NML_MODS**

*Note:* MODS *is a vector of type (class)* input_mod, used to store time dependent input data.

| | | | |
|---|---|---|---|
| | INTEGER | MODS(i)%MODE | **0**/1 If 0, parametric function is not used. If 1, replace input by Parametric function, and then SHIFT and MULTI have no effect as the same result can be achieved with MIN and MAX. |
| | INTEGER | MODS(i)%COL | **−1** Column number where the input data is read. −1 = data is not read from file. Column 1 is reserved for time. |
| | REAL | MODS(i)%MULTI | **1.0** Multiplies the value of a variable, when it is obtained by reading a from input file or by linking to another variable. Has no effect if Parametric function is used |
| | REAL | MODS(i)%SHIFT | **0.0** Shifts the variable value in the same units as defined in UNIT |
| | REAL | MODS(i)%MIN | Parameters for the parametric input function. These should be modified using the GUI, which shows a visual output of the parametric function. |
| | REAL | MODS(i)%MAX | |





| | | |
|---|---|---|
| REAL | MODS(i)%SIG | |
| REAL | MODS(i)%MJU | |
| REAL | MODS(i)%FV | |
| REAL | MODS(i)%PH | |
| REAL | MODS(i)%AM | |
| CHARACTER | MODS(i)%UNIT | **Unit of the input, depends on the variable:**<br>Temperature: `'K'`,`'C'`<br>Pressure: `'Pa'`,`'hPa'`,`'mbar'`,`'kPa'`,`'bar'`,`'atm'`<br>Relative humidity: `'%'`<br>Condensation sink: `'1/s'`<br>Short wave radiation: `'W/m2'`<br>Ion production rate: `'ip/cm3 s'`<br>Nucleation rate: `'1/cm3 s'`<br>Concentrations, emissions: `'#/cm3'`,`'ppm'`,`'ppb'`,`'ppt'`,`'ppq'` |
| CHARACTER | MODS(i)%TIED | If given, will link one variable with another instead of using the column number. |






## Appendix B: Output files, variables and folder structure

Figure B1 shows how the output directory names are formed from the date or index, Case and Run names, and what files are written in the output directory. The directories are automatically created except for the Common Out (`INOUT_DIR`). NetCDF files are binary files and must be read with a compatible software. After installing ARCA the user has the necessary Python

packages to access NetCDF (by 'import netCDF4'); other software includes ncdump, Octave, Panoply etc.

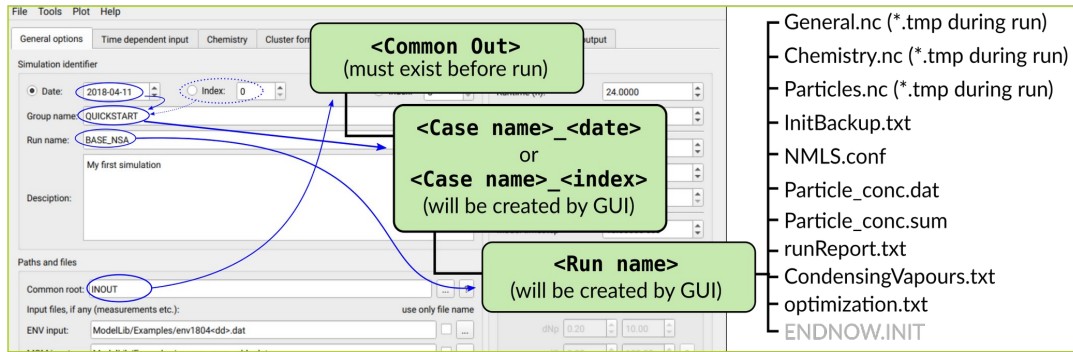

**Figure B1: Output directory naming and the files created in each run.**



## Appendix C: INITFILE

With INITFILE, the model can be run by loading it in the GUI: 1) by drag and drop 2) Ctrl-O or 3) 'Load settings', or run

from terminal by giving the file path as command line option:

```
./arcabox.exe path/to/INITFILE
```

**Example INITFILE, used in simulation discussed in section 4.2**

```
      # --------------------------------------------------
#                 ARCA box setting file #1
      #         Created at: Feb 23 2022, 13:08:10
      # --------------------------------------------------

      &NML_PATH
INOUT_DIR = 'INOUT'
       CASE_NAME = 'PAPER'
       RUN_NAME = 'FINAL_CW40'
      /

&NML_FLAG
       CHEMISTRY_FLAG = .TRUE.
       AEROSOL_FLAG = .TRUE.
       ACDC_SOLVE_SS = .FALSE.
       ACDC = .FALSE.
CONDENSATION = .TRUE.
       COAGULATION = .TRUE.
       DEPOSITION = .FALSE.
       CHEM_DEPOSITION = .TRUE.
       MODEL_H2SO4 = .FALSE.
ORG_NUCL = .FALSE.
       PRINT_ACDC = .FALSE.
       OPTIMIZE_DT = .TRUE.
       AFTER_CHEM_ON = .TRUE.
       AFTER_NUCL_ON = .FALSE.
630   /

      &NML_TIME
       RUNTIME = 6.0
       DT = 0.001
FSAVE_INTERVAL = 5
       PRINT_INTERVAL = 300
       FSAVE_DIVISION = 200
       DATE = ''
       INDEX = '0003'
640   /

      &NML_PARTICLE
       PSD_MODE = 1
       N_BINS_PAR = 100
MIN_PARTICLE_DIAM = 5e-9
       MAX_PARTICLE_DIAM = 10e-6
       N_MODAL = 6e3
       MMODAL_INPUT_INUSE = 1
       DMPS_FILE = ''
EXTRA_PARTICLES = ''
       MMODAL_INPUT = '95e-9 0.200 1'
       DMPS_READ_IN_TIME = 0.0
       DMPS_HIGHBAND_LOWER_LIMIT =
       DMPS_LOWBAND_UPPER_LIMIT =
USE_DMPS = .FALSE.
       USE_DMPS_PARTIAL = .FALSE.
      /

      &NML_ENV
ENV_FILE = '/****/****/03-Artikkelit/001-ARCA/Pathak2007/apine_profile.dat'
       SPECTRUMFILE = 'ModelLib/Photolyse/Spectra/swr_distribution.txt'
       SWR_IN_LOWER = 300
       SWR_IN_UPPER = 4000
       SWR_IS_ACTINICFLUX = .FALSE.
LOSSES_FILE = 'INOUT/PAPER_0000/WALLLOSS/losses.dat'
       CHAMBER_FLOOR_AREA = 5.0
       CHAMBER_HEIGHT = 2.0
       EDDYK = 0.05
       USTAR = 0.9
ALPHAWALL = 5e-05
       CW_EQV = 4e-05
      /

      &NML_MCM
MCM_FILE = '/****/****/03-Artikkelit/001-ARCA/Pathak2007/apine_profile.dat'
      /

      &NML_MODS
       MODS(1)  = 0 -1 1.00d+00 2.00d+01 1.00d+01  1.00d+05 2.340d0 12.000d0 0.000d0 0.000d0 1.000d0 'C' '' ! TEMPK
MODS(2)  = 0 -1 1.00d+00 1.00d+03 1.00d+01  1.00d+05 2.340d0 12.000d0 0.000d0 0.000d0 1.000d0 'hPa' '' ! PRESSURE
```



```
      MODS(3)   = 0 -1 1.00d+00 1.00d+01 1.00d+01  1.00d+05 2.340d0 12.000d0 0.000d0 0.000d0 1.000d0 '#' '' ! REL_HUMIDITY
      MODS(7)   = 1 -1 1.00d+00 0.00d+00 9.50d-01  2.00d+00 0.840d0 -0.28572d0 2.36d0 -0.30d0 0.65d0 '#' '' ! ION_PROD_RATE
      MODS(16)  = 0  6 1.00d+00 0.00d+00 1.00d+01  1.00d+05 2.340d0 12.000d0 0.000d0 0.000d0 1.000d0 'ppb' '' ! O3
      MODS(24)  = 0 -1 1.00d+00 8.00d+01 1.00d+01  1.00d+05 2.340d0 12.000d0 0.000d0 0.000d0 1.000d0 'ppm' '' ! BUT2OL
MODS(151) = 0 -1 1.00d+00 3.83d+01 1.00d+01  1.00d+05 2.340d0 12.000d0 0.000d0 0.000d0 1.000d0 'ppb' '' ! APINENE
      /

      &NML_MISC
       LAT = 60.0
LON = -25.0
       WAIT_FOR = 0
       DESCRIPTION = 'Experiment 13 from: Pathak, R. K., C. O. Stanier, N. M. Donahue, and S. N. Pandis (2007), Ozonolysis of
      a-pinene at atmospherically relevant concentrations: Temperature dependence of aerosol mass fractions (yields), J.
      Geophys. Res., 112, D03201, doi:10.1029/2006JD007436.<br>Psat from EVAPORATION, Par loss rate constant from the
reference, enhanced in the beginning, Duration of Ozone concentration increase 60 seconds. Chemical wall loss from
      parametrization.'
       CH_ALBEDO = 0.2
       GR_SIZES = ''
      /

      &NML_VAP
       USE_ATOMS = .TRUE.
       VAP_NAMES = 'ModelLib/MTVapours_EVAP_LT_E-6.dat'
       VAP_ATOMS = 'ModelLib/elements_MTVapours_EVAP_LT_E-6.dat'
705   /

      &NML_ACDC
       ACDC_SYSTEMS = 0,0,0,0,0
       ACDC_links(1) = 'A H2SO4 N NH3'
ACDC_links(2) = 'A H2SO4 D DMA'
       ACDC_links(3) = 'A H2SO4 D DMA'
       ACDC_links(4) = 'A H2SO4 N NH3'
       ACDC_links(5) = 'A H2SO4 N NH3'
      /
      &NML_PRECISION
       DDIAM_RANGE = 0.100000,1.000000
       DPNUM_RANGE = 0.100000,2.000000
       DVAPO_RANGE = 0.100000,1.000000
720   /

      &NML_CUSTOM
       FLOAT_CONC_AFTER_HRS = 0
       ORGANIC_DENSITY = 1400
VP_MULTI = 1
       HARD_CORE_DENSITY = 1200
       DT_UPPER_LIMIT = 30,70,70
      /

&NML_FLAG
       FILE_TIME_UNIT = 'sec'
      /

      # Following settings are for the GUI and not directly used by the model -----
# RAW_INPUT = &NML_FLAG<br> FILE_TIME_UNIT = 'sec'<br>/
      #  INPUT_SETTINGS  =  'env_file:/****/****/03-Artikkelit/001-ARCA/Pathak2007/apine_profile.dat  mcm_file:/****/****/03-
      Artikkelit/001-ARCA/Pathak2007/apine_profile.dat           dmps_file:              extra_particles:
      losses_file:INOUT/PAPER_0000/WALLLOSS/losses.dat      spectralFunctions:ModelLib/Photolyse/Spectra/swr_distribution.txt
      stripRoot_env:.FALSE. stripRoot_mcm:.FALSE. stripRoot_par:.FALSE. stripRoot_xtr:.FALSE.'
# BATCH_SETTINGS = 'batchRangeDayBegin:2020-01-01 batchRangeDayEnd:2020-01-05 batchRangeIndBegin:0 batchRangeIndEnd:0
      indexRadioIndex:.TRUE. indexRadioDate:.FALSE. createBashFile:.TRUE. batchRangeDay:.FALSE. batchRangeInd:.TRUE.'
      #-----------------------------------------------------------------------
```





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
