# Peer review of "Atmospherically relevant chemistry and aerosol box model – ARCA box (version 1.2)"

_Geoscientific Model Development, 2022_

## Author Comment (AC1)

Dear Editor and Referees

First, we want to thank you for your time and comments to point out aspects which should be clarified or added to improve the quality of this manuscript. Below we provide point-by-point replies to your comments. Our replies are written in blue and new text that will be added to the manuscript is written in red.

In light of the minor updates that have been done in the software, the version is now 1.2.2. and the title would need to be edited to reflect this. However, as the third number is a minor one, we decided to leave only the major version in the title and it is changed to:

Atmospherically relevant chemistry and aerosol box model – ARCA box (version 1.2)

We still refer to version 1.2.2. in the abstract and in the introduction.

**Referee # 1:**

1) At the beginning of the introduction, a very short overview on possible applications for aerosol and chemical box models is presented. However, the examples given are solely limited to the published work of the authors. The introduction would greatly benefit when discussing the ARCA model in the context of other aerosol and chemical box models in a technical and historical context. Especially, focusing on models and publications outside the authors' group.

Thank you for this comment. The focus of the introduction was not to give an overview of the current state of process modelling. That said, we agree that some expansion on the topic would benefit the reader. We have also made it more clear, which previous models of the authors precede ARCA.

We have replaced the first paragraph of the introduction with the following two paragraphs:

Aerosol and chemical models can be categorized by their dimensionality, where zero-dimensional models are called box models. However, a box model is generally used as a core module in dimensional models (column, regional or global models), and therefore it makes sense to further describe a model by its complexity and level of details in the chemistry and physics. In this respect, aerosol models can be divided sectional or modal approach, where the first uses discretized size representation and the latter treats the aerosol population as a combination of modes (Zhang et al. 1999). The modal approach is often used in global models due to to computational efficiency (e.g. the M7, Vignati et al. 2004), but as the sectional approach can be kept coarse enough, models such as SALSA (Kokkola et al. 2018) have found their place in large scale use as well as smog chambers. Here we turn our focus to sectional models with detailed chemistry and aerosol processes, which find their use in smog chamber, flow tube or atmospheric conditions. During the last decades of atmospheric research, several numerical process models for simulating gas and particle phase chemistry and dynamics have been developed. Examples from recent years include (but are not limited to) KinSim, used in simulating the chemical evolution of various gas phase species aimed at studying indoor-air chemistry (Peng and Jimenez, 2019), MAFOR (Multicomponent Aerosol FORmation model, Karl et al. 2022), also a community aerosol dynamics model which also includes multiphase chemistry. Often models are focusing on some particular aspect of aerosol dynamics, such as nanoparticle chemistry and dynamics, e.g. MABNAG, (Yli-Juuti et al. 2013) and TOMAS and its extension SOM-TOMAS (e.g. Adams and Seinfeld 2002; Akherati et al. 2020),

particle chemistry and fine structure (e.g. KM-SUB, Shiraiwa et al. 2010) or cloud droplet chemistry and processes (CLEPS, Rose et al. 2018). A review paper by Smith et al. (2021) discusses the current status of the understanding of nanoscale aerosol chemistry, and also gives a good overview of the many process models that are used to solve aerosol and atmospheric chemistry and dynamics.

The history of ARCA box starts from the four models used by the authors: MALTE-box (Boy et al. 2006; Boy et al. 2011), ADCHEM (Roldin et al. 2011; Roldin et al. 2014), ADiC (Pichelstorfer and Hofmann 2015) and ACDC (Atmospheric Cluster Dynamics Code, McGrath et al. 2012; Ortega et al. 2012). These models have been used to study ambient phenomena, test and study complex chemical schemes or specialized applications such as molecular clustering, chemistry and deposition in lungs (e.g., Olenius et al. 2013; Myllys et al. 2019); Boy et al. 2013; Pichelstorfer and Hofmann 2015; Xavier et al. 2019; Pichelstorfer et al. 2021). The current work is the amalgamation of these models, with the aim of further development as an open-source community model.

2) As the authors clearly state (Sect. 1.1), the model in its current state neglects important processes for aerosols (e.g., aqueous-phase chemistry). However, the targeted user group is scientists that have little to no experience in aerosol and chemical modelling. This user group would greatly benefit from a section discussing the implications of the missing processes on the model's results (e.g., before section 6).

We agree that this discussion could be helpful. The question is of course a very difficult one to answer in the scope of this paper, this is an area of research which is very much developing at the moment, new processes and approaches to solve the multi-phase chemistry are being constantly developed and tested, and the overall picture is still an elusive one.

We have renamed section 6:

**6 Current limitations and future developments**

And edited the following section in the following way (grey is unchanged text):

This information will then be transferred to the condensation/evaporation module which will use the analytical predictor of dissolution (APD) method (Jacobson 1997a) to solve the gas-particle partitioning. The omission of particle phase chemistry (organic and inorganic) will affect the suitability of the model in marine air, where a considerable fraction of the aerosol mass is formed through oxidation processes which take place in aqueous phase (Xavier et al. 2022), and with very low concentrations of organic compounds the growth in the model would be almost solely based on the irreversible condensation of sulfuric acid which is formed in the gas phase chemistry, thus underestimating the total particle mass. Another major constituents of the secondary aerosol mass is nitric acid and ammonium, and these concentrations depend on the water content of the particles. One way to quantify in very broad strokes what a purely gas phase model is at worst missing would be to look at the measured aerosol composition around the world. Zhang et al. (2007) analysed the submicron aerosol AMS data from 37 field campaigns and found that organic compounds constitute on average 45 % (18–70%), sulfates 32% (10–67%), nitrates 10% (1.2–28%), ammonium 13% (6.9–19%) and chloride 0.6% (<D.L.-4.8%); the ranges in parenthesis are the range between different measurement locations. The current processes in the model are capable of bringing organic molecules, organic nitrates and sulfates (in the form of $H_2SO_4$ from $SO_2$ oxidation with OH) to the particle phase, but they would miss ammonium and chlorides. Some fraction of the nitrates and sulfates (and possibly organics too) are resulting from particle phase reactions in either aqueous

form or oligomerization. The effect of particle phase oxidation of DMS and the subsequent formation of methanesulfonic acid (MSA) is one example of a process which is relevant over the oceans (de Jonge et al. 2021), but not necessarily a crucial omission over boreal forest, where similar approaches as in ARCA have been in good agreement with the measured particle mass and size distributions (Roldin et al. 2019).

Another area of improvement is the addition of charged particles, which is a significant factor in chamber wall losses, as was already discussed in Sect. 4.3.

3) The authors claim that the GUI design is one of the main aspects that distinguishes ARCA from other box models. However, very little information on the GUI design is presented and is mainly limited to a series of figures from ARCA's GUI with very little information about it in the text (Sect. 3). Further elaboration on the GUI design is necessary in this section. This information should at least contain but is not limited to: (1) the GUI design principles, and (2) a workflow diagram and description of the GUI design/use including dependencies between the different tabs. At multiple locations in the text, the authors discuss possible errors. Since the targeted user group is expected to have little experience with box modelling, a proper error management is key to avoid misconfiguration followed by potentially wrong conclusion. An elaboration on the error message management should be added.

It has been a deliberate choice in this paper to focus on the underlying equations and description of processes in the model, and not intend it as a practical guide for setting up the model. This is because we have prepared the online user manual with instructions of workflow, with examples in text and videos, instructions with diagrams on workflows regarding specific details (such as creating a new chemistry scheme from scratch, batch processing, sensitivity studies etc.). The user manual, particularly on the online platform, is a medium where both the format and the wording can be more instructional and informal, and it is easy to update with new features or more in-depth instructions if needed. Writing the same information in a journal article would still leave the reader wanting for practical advice, or turn that section of the manuscript into very technical text to which the reader who doesn't have the program in front of them would find it very hard to relate in any practical way. After all, learning a program is best done by using it, and here the installed model with the online manual is the perfect combination.

That said, we have elaborated extensively on the design aspects as well as on the screenshots provided, and connected them to the example case that was used to produce Fig 10. In the process we chose to replace Fig 5b, 6a and 6b, as well as update 5a with the updated GUI. Also references to error management and general workflow are given. Still, we emphasize that the proper place for user guidance is the user manual, which can be written in more colloquial and informal way, thus providing much better pedagogical environment. This is also clarified in the revised manuscript.

Section 3 is edited accordingly:

 Therefore, we will here introduce the GUI only broadly with a general notion of one possible workflow, and emphasize that the user who wishes to learn to use the model, should rely on the online user manual. The manual allows the use of much more pictures, videos, interlinking between content and more informal addressing, thus forming a vastly better pedagogical platform for learning than this paper could do.

The added text begins after section 3:

**3.1 GUI design principles**

The primary principle of the GUI design has been that the numerical model is kept autonomous of the user interface. This ensures that the model can be run on as many systems as possible, even when the user has no access to a graphical OS, such as when working on an SSH connection to a HPC. Another aspect is that Fortran is a much more conservative language than Python and it has therefore better record of working reliably as intended on each OS. While Python is cross-platform, it's reliance on multiple extra packages and their rather quick updating schedule makes it more susceptible to introduce unintended behaviour in the GUI. While such issuesare usually fixed in the ARCA repository quite swiftly, it is highly convenient that the underlying numerical model works independently of the GUI. The independence between the model and the GUI is ensured throughthe INITFILE, where the GUI reads and writes a Fortran Namelist text file, which is used to initialize the numerical model.

Another design principle has been that the GUI should allow further model development, that is, it should be able to write, save and read in any custom options, even if they are not yet fully incorporated in the graphical layout. Some of these options might be later incorporated, some may turn out to be unnecessary. This principle manifests itself in the GUI's tab *Custom options*.

Third design principle has been to enable flexibility in the input to the model. All input is done via human readable text files. Not all modellers have their input data available in same units or time resolution, and the GUI lets the user to define theirs. However, this flexibility has limits and in the end the input files are expected to follow a certain structure which is covered in detail in the manual. Also the example files in the ModelLib directory can be used as a guide.

The last design principle is similar to what any graphical user interface typically aims to do: to collect the options of a program in meaningful groups and hierarchy, showing relevant and hiding unnecessary options, deriving information from user input such as parsing pathnames or calculating values, and provide assistance through help links or tool tips. It can be noted that while the amount of GUI options might seem intimidating, many of them are related to the way the input data is structured, and usually after the initial set up – which takes a while for the first time – the user can leave most options as they are and concentrate on testing further with only a few key options.

**3.2 Selected features of the GUI**

To give an overview of the GUI, figures 5–7 show selected screenshots of the GUI window. For in-depth instructions and descriptions, the user is directed to the online manual, which contains an instructional video and transcript of setting up a simulation with the provided example files. To keep this work concise, here we give a general overview of the key points in setting up a simulation. As an example, we have chosen the case which produces Fig. 10, and whose complete input is included in the directory ModelLib/Examples/APINENE_OXY. Figure 5a shows the tab *General options,* which contains many of the basic options for the simulation: naming of output, paths to input and output files, model time step and toggles for the main modules. Based on the choices in the "Modules in use", other tabs will be enabled or disabled, and the work procedure is to fill out the "forms" one tab at a time, and then use "Run ARCA".

After *General options*, the most crucial set of options are in the next tab *Time dependent input* (Figure 5b). Here the concentrations of precursors and many environmental variables are defined. The logic in this tab is that any other input than temperature and pressure are optional. Any variable which is not defined here as an input, will have a constant 0 value. One picks the variables that are given as input to the model from the right-hand panel "Available input variables" and moves them to selected list of input variables. Next, there are several options how to define the values of any variable. For example $O_3$ could come from measurements, in which case it would be read in from the ENV file from the column number defined in "Column/Link", or it could be a constant, in

which case it would be sufficient to "Shift" the default value 0 by the constant amount, or we might want to multiply the measured values by a constant "Multi". Another option would be to link the concentration to some other variable, and then one would use that variable name in "Column/Link". Finally, there is a way to use parametric input in the next tab "Parametric input", where a set of sliders can be used to create a smooth time series for any given input variable. This part of setting up the model is usually somewhat time consuming, but once properly done, the user can easily perform sensitivity tests by using the "Shift" and "Multi" options of any variable. Furthermore, the tool "Variations" can be used for sensitivity tests, which creates a batch of simulations where any given set of variables are varied within given ranges, containing all combinations of the different input settings.

Figure 6a shows the options related to aerosol module, such as size range and resolution of the PSD grid, its initialization with either measurements from file or using (multi)modal lognormal distribution (as is done in this case), and duration and size range of initialization. This tab is also where the pure liquid saturation vapour pressure properties are defined in the "Vapour file" and the optional "Vapour elemental composition" file. In our example case, a single mode log-normal particle size distribution was used to initialize the particles in the simulation. The (multi)modal PSD is defined by total particle number, mode diameter, standard deviation and relative contribution to the total PN. The GUI plots the resulting mode in real-time and calculates total PM and particle area, as these are sometimes reported and can be used to fine-tune an unknown modal parameter such as the mode width.

Figure 6b shows an example of the model output during the runtime, when it is used from the GUI. The preferred way of setting up the model the first time is to start the model from the tab *Run ARCA* by pressing "Run model with current settings" and dealing with the eventual error messages, which might appear due to misconfiguration. If the model enters the main loop (after printing "Starting main loop"), it is advisable to press "Force stop" and read the initial report of the model that is printed in the "Monitor" window. If there are no warning messages, and the reported unit conversions and other reported behaviour correspond to the intent of the user, the simulation can be performed in full.

Fig. 7 shows examples of the output plotting tools such as surface plots (panel a) and particle mass and size distribution plots (panel b). A useful tool is the *Live update* option, which shows the evolving particle size distribution surface plot during the simulation.

The captions of Figs. 5–7 have been updated accordingly.

4) The verification of the model (Sect. 4) is poorly presented and e.g. in Sect. 4.3, it is limited to just presenting two plots. Some discussion of the results should be added.

The model which we present in this paper is based on theories and modules which have all been previously published and peer-reviewed, and to verify that the model is working as intended, we have shown examples from the crucial calculations from individual modules, as this is in our view the best way to ensure that there are no bugs in the functions and subroutines. Based on these we are confident that the modules work in the way they were originally intended to work.

The numbering of sections 4.2 and 4.3 were mixed (this is fixed), and we assume this comment refers to the section which currently is "4.3 Verification of the coagulation module". There are two plots but there is also text discussing them. We have elaborated the discussion a bit further for both sections 4.2 and 4.3:

To section 4.2 (in the current, corrected numbering) we have added a sentence in the end:

Fig. 10 (n.b. this used to be Fig. 9) shows that on one hand the coagulation kernel is calculated in the same way as in the source, but also that the coagulation losses are correctly applied, and therefore the temporal evolution of the discrete particle size distribution in ARCA is identical with an analytical solution of the same initial size distribution.

In Section 4.3 new text is added, for example a sensitivity test with chemical wall loss parametrizations. Below is the is reformulated section (red is new or edited text):

**4.3 Chemistry, condensation and loss routines**

We simulated α-pinene oxidation in high ozone and low OH chamber environment, similar as in experiment 13 in Pathak et al. (2007) (Figure 11). The 10 m$^3$ Teflon bag chamber was initially filled with seed particles, α-pinene and 2-butanol as OH scavenger. Then O$_3$ was introduced to the chamber. The reported mean particle loss rate (0.3/h) was expanded to size and time dependant profiles using Pierce et al. (2008), who measured the losses in the same chamber in more detail. They report an initially higher rate of loss in all their experiments, and attribute this to the growth of the particles (leading to slower average  loss rate) and in the enhanced loss rate of initially charged seed particles. Since ARCA in its current form does not consider charge effects, we chose to use the reported loss rates, with higher initial rate. It means that the simulation is not showing the effect of the loss rate calculated by the particle loss module, but it does still verify that the rates are properly applied – given the good agreement in total particle mass time series. The simulation used moderate wall loss of condensing vapours (ALPHAWALL = 5×10$^{-5}$, CW_EQV = 40 µm/m³ and EDDYK = 0.05 s$^{-1}$). The final SOA mass yield of 0.169 (final SOA mass / initial α-pinene mass) calculated by ARCA agrees well with the reported 0.17. Without evaporation from the particles, invoked by the accumulation of vapours to walls and the consequential supersaturation decrease, the calculated yield is 0.176. The final concentration of vapours deposited on the walls amounted to 13.4 µg/m³. In the chemistry scheme built for this test, ozone concentration was set to the target (250 ppb) at time 1105 seconds, and was then let evolve according to the chemical reactions. Without chemical wall loss the simulated yield is 0.286, overshooting the reported yield by a factor of 1.7. The pure liquid saturation vapour pressures used were derived using the EVAPORATION method. The chemistry was acquired from MCM and amended with the PRAM, and $p_{sat}$ data was downloaded from UmanSysProp (umansysprop.seaes.manchester.ac.uk, Sect. 2.5), using the tools in ARCA box. The ARCA distribution contains an example case which produces similar simulation than was used to produce Fig. 11. Here the initial time without ozone is omitted and the simulation starts immediately with 250 ppb concentration. To accommodate for the initial particle losses, the initial particle number concentration was set to 4537 cm$^{-3}$. This simplification makes configuring the model much more straightforward, and produces the same result apart from the shift in time when compared with the experiment data.

[Figure]

**Figure 11: Comparison against α-pinene oxidation experiment (exp. 13, Pathak et al. 2007). Size-resolved aerosol wall losses were measured for the same chamber in Pierce et al. (2008). Moderate chemical wall loss is calculated with parametrisation from ARCA box using `ALPHAWALL` = 5×10⁻⁵, `CW_EQV` = 40 μm/m³ and `EDDYK` = 0.05 s⁻¹.**

We also performed sensitivity runs where we tested the vapour wall parameters $C_w$ and $\alpha_w$. Changing $C_w$ an order of magnitude lower and higher (to 4 and 400 μm/m³) changed the resulting yield to 0.179 and 0.166, and final total vapour mass on the wall was 7.16 and 16.46 μg/m³, respectively. For reasons discussed in section 2.7 about the limit of very small $\alpha_w$, we saw no practical difference in the yield or vapour mass lost to walls after $\alpha_w > 10^{-4}$, where the uptake apparently has become diffusion-limited. The selected default value of $\alpha_w$ (5×10⁻⁵) can be considered to be close to the value where the surface reactions start playing a role on the uptake.

**Table 1: Sensitivity of the vapour wall losses to parameters $C_w$ and $\alpha_w$.**

| | $C_w$ [μmol/m³] | $\alpha_w$ [-] | Final yield [-] | Vapour concentration on walls [μg/m³] |
|---|---|---|---|---|
| **Base case** | **40** | **5×10⁻⁵** | **0.169** | **13.4** |
| $C_w \times 10$ | 400 | 5×10⁻⁵ | 0.166 | 16.5 |
| $C_w \times 0.1$ | 4 | 5×10⁻⁵ | 0.179 | 7.2 |
| $\alpha_w \times 0.1$ | 40 | 5×10⁻⁶ | 0.196 | 13.5 |
| $\alpha_w \times 10$ | 40 | 5×10⁻⁴ | 0.164 | 13.3 |
| $\alpha_w \times 100$ | 40 | 5×10⁻³ | 0.164 | 13.4 |

5) Before the authors provide a revised version of the manuscript, they should perform an editorial check focusing on the consisted use of the Copernicus rules (e.g., use of figure within the text), correct numbering of sections, etc.

Thank you for pointing out the mistake in section numbering, we have fixed the numbering of sections 4.3 and 4.2. We also unified the labelling of the figures from capital A and B to (a) and (b). Unfortunately, we are not sure what is meant here by the problems with figures within text, maybe this is something which can be corrected at the (eventual) final preparation?

Specific comments:

L20: Stating that "any compatible chemistry scheme" is ambiguous since currently the model is limited to gas-phase chemistry.

We have added "gas phase" in the sentence: "The **gas phase** chemistry module incorporates the Master Chemical Mechanism…".

L109: Is the user notified of this if the time step is selected too large by the user? Is there an upper limit of the time step length?

The optimal (and too large) time step always depends on the dynamic conditions. The GUI allows the user to input at most 99 seconds time step, but the model itself accepts any value. The default (10 seconds) value is a good compromise between dynamic stability and computational efficiency for most simulations. However, the concerned user should use the optimized time step, especially if the conditions are very variable, as they would be in a batch mode chamber simulation. We have added the following sentence to the end of this paragraph:

The default time step (10 seconds) is usually a good compromise between stability and computational efficiency, but the model also includes a time step optimization algorithm (described in Sect. 1.3).

Figure 1: The definition of PSD is missing in the caption.

Chanced "PSD" to "particles"

L271: The last sentence seems to be out of place. Shouldn't this be specified in the introduction?

Absolutely correct, the paragraph was reformulated by adding in the beginning: "The default ARCA has slots for five ACDC modules, which can be switched on and off individually." and in the end: " Furthermore, the number of slots for ACDC modules can be increased from five with minor code modifications.".

L285: Is the user made aware (e.g., in the GUI) that he is responsible to select the PSD size range? Is an error raised if he fails to do so? This could be used by the authors as an example for the error management (see general comment 3).

The GUI has a section "Aerosol structure" in the tab "Aerosol", where the number of bins and size range is selected, with default values for the diameter range 1.07 nm − 2µm. The GUI will issue an error if the size is deliberately left empty (or not a number).

L397: What is your justification to select the same value for all compounds? Can the user change this value for single compounds? This would make it easier to tailor ARCA to individual chambers.

The only justification is that these values are not known for individual components, and especially with large number of vapours having them set individually would introduce a huge number of free parameters. Instead we have relied on previous experiments, who use a constant value of 1e-5/s (Matsunaga and Ziemann, 2010; Zhang et al., 2014).
However, there could well be good reasons for setting the values for some compounds or groups separately, and we have now updated the model's input module so that both surface tension and $\alpha_w$ can optionally be set in the "Vapour"-file (which primarily defines compound-specific saturation vapour pressure) for each component, if the user so wishes. We have amended the sentence on line 397 as follows:  however, these are generally not known, and a constant value of $10^{-5}$ is used as a first estimate, following e.g. Matsunaga and Ziemann (2010) and Zhang et al. (2014). Similar to surface tension data, individual $\alpha_w$ (or correction factors to constant ALPHAWALL if set to negative) can optionally be set in the file which defines the pure liquid saturation vapour pressure data.

In the end of the paragraph, some further elaboration is added, this will later be referred in section 4.3:

McMurry and Grosjean (1985) discussed the effect of $\alpha_w$. In the limit of very small $\alpha_w$, Eq. (10) can be reduced to  $k_{gas \to wall, q} = \frac{A}{V} \frac{\alpha_w \bar{v}}{4}$  and the uptake of gases is then limited mostly by surface reactions. With increasing $\alpha_w$, the uptake becomes diffusion-limited, finally reducing to  $k_{gas \to wall, q} = \frac{2A}{\pi V} \sqrt{k_e D}$ , the separating value for $\alpha_w$ which divides the two modes of uptake being at  $\alpha_w = \frac{8}{\pi} \frac{\sqrt{k_e D}}{\bar{v}}$ .

L432: The online user manual is nice. Adding a link to it here would be helpful.

Absolutely, link added.

L439: This statement is ambiguous, unspecific, and misleading. What distinguishes "good" and "better" science? I suggest rephrasing this statement focusing e.g. the avoidance of systematical errors when performing simulations or similar.

This sentence was meant more as a comic relief before the examples why the GUI makes the model easier to configure and reconfigure. In light of this comment it might be best to remove it completely and instead focus on the actual practical aspects.

Section 5.3: In addition to the model code, please provide the modelling data of the simulations used in the verification section (e.g., using Zenodo).

This is now done along with the updated code, and can be found in the ModelLib/Examples directory. The user can load the INITFILE and the case is then available.

Section 6: In Sect. 1.1, you state that organic liquid phase chemistry is not considered in ARCA but may be important (e.g., on the thermodynamics, formation of secondary organic aerosols). Is there a reason why you currently do not plan to include this chemistry in future model versions? In the case when using MCM, the coupling to CLEPS (Mouchel-Vallon et al., 2017) should be strait forward.

This is definitely a good suggestion for the upcoming work. As stated in the paper, the current work is concentrated on adding particle phase chemistry to the model. CLEPS seems promising and is something which we will definitely consider. At the moment the model's aerosol module is lacking

the necessary code for particle phase chemistry and the corresponding coupling with the chemistry module, and therefore adding CLEPS to a discrete PDS at this point is not straightforward.

**Referee #2:**

The idea of using different timesteps for different processes is interesting. I am wondering if the authors can say how much computational time is saved compared, for example, to the fixed timestep mentioned on line 121. I imagine this is related to the chemical conditions used, but it would be good to have a ballpark idea. One thing that is not clear to me is whether the possibility of using different timesteps for different processes is a consequence of the fact that the processes are run in a sequence (line 105). As a side note to this point, it would be good if the authors can comment on how the model ensures that mass is conserved when "moving" from one process/module (e.g. gas phase chemistry) to the next in the sequence (e.g. particle formation).

Yes, the computation time saved with the time step optimization depends a lot on the conditions of the simulation, processes that are used and the precision parameters. As a ballpark estimate the optimized time step can be 2–3 times faster, but on the other hand it can be even an order of magnitude slower. This is largely because the default 10 s time step is already quite large for dynamic conditions and will already lead to relatively large changes of concentrations (~10% < dC < ~50%) for isolated compounds, especially in the condensation module. For this reason we didn't quantify the time differences between the two modes. Based on this comment we have added the following sentence before the last sentence of section 1.3:

"With loose tolerances (5–10%) the simulation time is usually shortened, while tight tolerances (0.2–3%) will increase runtime markedly."

The time step optimization is possible because the modules are run in a sequence.

About the mass conservation, all fluxes in the model are mass conserving – with one exception. New particle formation will produce ~1.1 nm particles whose composition is assigned to "GENERAL", and the participating monomer concentrations are unaffected. As the stable molecular cluster concentrations are always several orders of magnitude smaller than monomer concentrations, this error can be considered negligible.

On line 94 it is said that all particles are considered to be liquid droplets, but on line 98 it is said they are void of water. This sounds like a contradiction. Is is also not clear to me what is the difference between the aerosol module mentioned in this paragraph (lines 94-99) and the ACDC module. The following sentence (lines 100-101) implies they are both used in the model, but the text on page 2 (lines 73-76) seems to suggest that the model uses the ACDC only. The APC scheme is also mentioned on page 2, but it is not clear whether it is part of the ARCA model or not.

Here liquid does not imply water but rather phase. The composition of the particles consists of organic compounds and possibly of sulfuric acid, and this mixture of solvents is considered liquid (there is no mention of water). The implication of this assumption is that the droplets would assume spherical form upon coagulation and that the components are thoroughly mixed. We have refined the sentence to: "...spherical, liquid-**phase** droplets with…".

The aerosol module simulates the time evolution of the aerosol particle size distribution from ~1.1 nm upward, while the ACDC module simulates the formation of the very initial particles below the size range covered by the aerosol module. While larger particles are modelled by a sectional framework, modelling the initial nucleation requires molecule-by-molecule treatment, which is implemented by the ACDC application in ARCA.

As mentioned on page 2, APC is used in the model, and later section 2.5 describes the module and APC in detail, so it should be clear that APC is used to calculate condensation of organic compounds. To clarify further, we have added the following words in the first sentence of section 1.2: "...4) gas-particle partitioning (condensation and evaporation, **using the APC scheme**) 5) coagulation..."

I think there should be a little more explanation about the wall loss process described in section 2.7 (and also about the "gas-wall partitioning" in Figure 1. What is it meant for? Indoor studies and/or environmental chambers? Is it inactive in ambient studies?

As stated in the beginning of 1.2, any of the main processes in Fig 1 (yellow boxes) can be switched on or off as the user sees necessary. As suggested by the comment, the reversible vapour loss would be useful for indoor studies or chamber studies, but not in ambient studies. It is up to the user to switch the losses on/off depending on the simulation.

We updated Fig 1, the gas-wall partitioning is now called with a more appropriate name "reversible vapour losses". We have also expanded section 2.7 on several locations (this section was also discussed in the other reviewer comment). To best get idea of the editions, the whole Sect. 2.7 is included here, with unchanged text as grey, red showing additions or editing:

In chamber experiments, losses of gases and particles to the chamber walls have a large influence on the outcome – same is true for simulations. In ambient conditions losses are more complex and harder to measure, due to advection, vertical mixing and different deposition processes. For simulations in confined spaces such as in a reaction chamber, the losses can be parameterized to some extent. ARCA considers losses of gas phase compounds and particles in separate steps, as shown in Fig. 1. The compounds considered in the vapour wall loss module are the same set of organic compounds that are condensing on the particles. The wall loss is based on the theory proposed by McMurry and Grosjean (1985), and is a reversible process characterized by two reaction rates $k_{gas \to wall}$ and $k_{wall \to gas}$

$$\frac{dC_{q,g}}{dt} = -k_{gas \to wall} C_{q,g} + k_{wall \to gas} C_{q,w},$$

$$\frac{dC_{q,w}}{dt} = k_{gas \to wall} C_{q,g} - k_{wall \to gas} C_{q,w}$$

(1)

[revised manuscript text omitted]

New figure (Fig. 5) is added:

[Figure]

**Figure 5: Particle loss rates as a function of particle diameter, friction velocity (u*) and chamber volume (given as floor area × height) as calculated by the aerosol loss rate parametrization. The rates are calculated at room temperature (20°C) and pressure (1 atm).**

line 81: delete "using"

deleted

line 99: instead of "restrictions" I suggest "limitations"

changed to "limitations"

figure 1: I suggest changing "wall loss of particles" to "particle losses" for consistency with table 1 and related text (or viceversa, change the table).

Yes, good point. Changed Fig. 1 accordingly. Since particle losses can be given as rates they can be used in a more general way than wall losses.

line 226: can you clarify what is meant by "numerical diffusion"?

The fixed grid particle size distribution method is unable to handle continuous growth, instead particles are always redistributed between discrete size bins, therefore introducing particles in sizes where there actually are none. For example, when particles in bin $k$ grow even just a little due to condensation, to conserve both mass and total number concentration a fraction of particles from bin $k$ must be moved to bin $k+1$. These particles will in turn in the next time step grow again by a tiny amount, and now a fraction from $k+1$ is moving to bin $k+2$. This is called numerical diffusion, since it resembles physical diffusion but is completely artificial.

We have edited the sentence accordingly:

The FS method is very robust during computation but **is not able to handle continuous growth and therefore** suffers from numerical diffusion **(Jacobson 1997b). This is the result of the redistribution, where particles are assigned to bins below and above the actual size the particles have grown (or shrank), thereby introducing particles that are larger (or smaller) than the condensation or evaporation produces. In time this will lead to spreading of the size distribution, much like a diffusive process.**

And furthermore added the sentence in the end of this paragraph:

Both PSD methods used conserve mass and total particle number.
The reference (this reference was already used elsewhere):
Jacobson, M. Z. (1997a). Development and application of a new air pollution modeling system—II. Aerosol module structure and design. *Atmos. Environ.*, 31, 131-144, doi:10.1016/1352-2310(96)00202-6.

line 271: it is said here that "ARCA has slots for five ACDC modules". Does it means that there are 5 aerosol bins, as default? Related to this, does the ARCA GUI allow modification of the ACDC settings or one would need to modify those separately, following the ACDC user manual?

This means that there can be at most 5 ACDC molecular cluster chemistries working simultaneously (this number can be increased with minor code editing). Each ACDC cluster system produces a formation rate of new particles formed from the given vapor species, and in the end of the time step

these rates are summed together. The summed formation rate is used to introduce new particles in the first bins of the discrete aerosol structure, which can have any number of bins (this number would be defined by the user in the GUI, and is 100 by default).

There is seldom need for using 5 different ACDC modules simultaneously, and in practice the modules can be used to easily switch (with a check box in the GUI) between similar cluster systems with different input (as is now done for H2SO4-NH3 and H2SO4-DMA which are calculated with two different sets of theoretical input data for these species), without the need to create new ACDC systems or recompile ARCA.

For the advanced user, the GUI allows rebuilding the ACDC cluster systems from the three definition files: the "SYSCONF" which defines all the cluster compositions on a molecular level; the "ENERGIES" which gives the enthalpy (dH) and entropy (dS) of the clusters and "DIPOLES" which defines the dipole moments. Of these files only SYSCONF is such where manual editing makes any sense, as the energies and dipole moments are usually the result of extensive quantum-chemical calculations. In the GUI the user can change the paths to these files (by using the GUI's editing tool), run the separate standalone ACDC main program to update the ACDC Fortran source files and recompile ARCA to update the main model, all with a couple of button pushes.

The repository containing ACDC has the aforementioned data files for many cluster systems, and many research groups produce their own. The ACDC manual is very extensive, and the ARCA manual has been concentrated on the steps needed to update any of the ARCA's ACDC chemical systems, if the user wants to test different input data for nucleation.

line 511: add link/reference to "UmanSysProp" and say what it is.

UmanSysProp is first introduced earlier in section 2.5, so we will add a reference to that section. Furthermore, we added a sentence in sect. 2.5 describing the UmanSysProp in their own words:

ARCA includes a tool that can be used to extract the temperature dependent $p_{sat}$ information using the methods in the UmanSysProp (http://umansysprop.seaes.manchester.ac.uk)**, an online facility for calculating the properties of individual organic molecules, currently hosted at the University of Manchester**.

Added (see section 2.5) in line 511

line 517: why do you say KPP is not necessary for ARCA? That seems to contradict the statements in section 2.1 and 2.1.2. The VODE (or DVODE? Is it the same thing?) solver is mentioned only once earlier in the paper. Please clarify if it is VODE or KPP (or perhaps both?) that are used to solve the differential equations system.

KPP is not strictly necessary, because here we are talking about the software KPP, whereas earlier we talk about the *code produced* by KPP. If the user uses the example chemistry schemes, there is no need to have software KPP to produce code for chemistry. Of course in practice installing KPP enables one to produce the necessary source code for creating additional chemistry schemes.

About VODE/DVODE, we actually meant DVODE (which is the double precision version of VODE solver), and corrected this in the text. DVODE is used in ACDC and in the vapour wall loss module. KPP has a range of options for the solvers used, and in ARCA's chemistry tools the default setup is using the Rosenbrock solver (as mentioned in 2.1.2). To summarize, KPP is the software

that produces the source code to define and solve the system of differential equations, and in that code a solver is used (Rosenbrock by default, but there are others, as described in the KPP manual). This source code is used as a module in ARCA. Therefore, once the chemistry module source code is produced, KPP is not necessary to run ARCA.

In Section 5.2. Are there any libraries or dependencies that need to be installed besides the netcdf libraries? e.g Python graphic libraries for the GUI and plotting?

Yes, there are. We have added a sentence in the end of section 5.2:

The script will install the necessary Python packages (most importantly PyQt5, PyQtGraph, NumPy, Matplotlib and netCDF4) and compile the model.

**References used in this response**

Adams, P. J. & Seinfeld, J. H. (2002). Predicting global aerosol size distributions in general circulation models. *J Geophys Res*, 107, doi:10.1029/2001jd001010.

Akherati, A., He, Y., Coggon, M. M., Koss, A. R., Hodshire, A. L., Sekimoto, K., Warneke, C., de Gouw, J., Yee, L., Seinfeld, J. H., Onasch, T. B., Herndon, S. C., Knighton, W. B., Cappa, C. D., Kleeman, M. J., Lim, C. Y., Kroll, J. H., Pierce, J. R. & Jathar, S. H. (2020). Oxygenated Aromatic Compounds are Important Precursors of Secondary Organic Aerosol in Biomass-Burning Emissions. *Environmental Science &mathsemicolon Technology*, 54, 8568-8579, doi:10.1021/acs.est.0c01345.

Boy, M., Hellmuth, O., Korhonen, H., Nilsson, E. D., ReVelle, D., Turnipseed, A., Arnold, F. & Kulmala, M. (2006). MALTE - model to predict new aerosol formation in the lower troposphere. *Atmos Chem Phys*, 6, 4499-4517, doi:10.5194/acp-6-4499-2006.

Boy, M., Mogensen, D., Smolander, S., Zhou, L., Nieminen, T., Paasonen, P., Plass-Dülmer, C., Sipilä, M., Petäjä, T., Mauldin, L., Berresheim, H. & Kulmala, M. (2013). Oxidation of $SO_2$ by stabilized Criegee intermediate (sCI) radicals as a crucial source for atmospheric sulfuric acid concentrations. *Atmos Chem Phys*, 13, 3865-3879, doi:10.5194/acp-13-3865-2013.

Boy, M., Sogachev, A., Lauros, J., Zhou, L., Guenther, A. & Smolander, S. (2011). SOSA – a new model to simulate the concentrations of organic vapours and sulphuric acid inside the ABL – Part 1: Model description and initial evaluation. *Atmos Chem Phys*, 11, 43-51, doi:10.5194/acp-11-43-2011.

Bruun, H. H. (1996). Hot-Wire Anemometry: Principles and Signal Analysis. *Meas. Sci. Technol.*, 7, doi:10.1088/0957-0233/7/10/024.

Jacobson, M. Z. (1997b). Development and application of a new air pollution modeling system—II. Aerosol module structure and design. *Atmos. Environ.*, 31, 131-144, doi:10.1016/1352-2310(96)00202-6.

Jacobson, M. Z. (1997a). Numerical Techniques to Solve Condensational and Dissolutional Growth Equations When Growth is Coupled to Reversible Reactions. *Aerosol Sci Tech*, 27, 491-498, doi:10.1080/02786829708965489.

de Jonge, R. W., Elm, J., Rosati, B., Christiansen, S., Hyttinen, N., Lüdemann, D., Bilde, M. & Roldin, P. (2021). Secondary aerosol formation from dimethyl sulfide – improved mechanistic understanding based on smog chamber experiments and modelling. *Atmos Chem Phys*, 21, 9955-9976, doi:10.5194/acp-21-9955-2021.

Karl, M., Pirjola, L., Grönholm, T., Kurppa, M., Anand, S., Zhang, X., Held, A., Sander, R., Maso, M. D., Topping, D., Jiang, S., Kangas, L. & Kukkonen, J. (2022). Description and evaluation of the community aerosol dynamics model MAFOR v2.0. *Geosci Model Dev*, 15, 3969-4026, doi:10.5194/gmd-15-3969-2022.

Kokkola, H., Kühn, T., Laakso, A., Bergman, T., Lehtinen, K. E. J., Mielonen, T., Arola, A., Stadtler, S., Korhonen, H., Ferrachat, S., Lohmann, U., Neubauer, D., Tegen, I., Drian, C. S.-L., Schultz, M. G., Bey, I., Stier, P., Daskalakis, N., Heald, C. L. & Romakkaniemi, S. (2018). SALSA2.0: The sectional aerosol module of the aerosol–chemistry–climate model ECHAM6.3.0-HAM2.3-MOZ1.0. *Geosci Model Dev*, 11, 3833-3863, doi:10.5194/gmd-11-3833-2018.

Lai, A. C. K. & Nazaroff, W. W. (2000). MODELING INDOOR PARTICLE DEPOSITION FROM TURBULENT FLOW ONTO SMOOTH SURFACES. *J Aerosol Sci*, 31, 463 - 476, doi:https://doi.org/10.1016/S0021-8502(99)00536-4.

Matsunaga, A. & Ziemann, P. J. (2010). Gas-Wall Partitioning of Organic Compounds in a Teflon Film Chamber and Potential Effects on Reaction Product and Aerosol Yield Measurements. *Aerosol Sci Tech*, 44, 881-892, doi:10.1080/02786826.2010.501044.

McGrath, M. J., Olenius, T., Ortega, I. K., Loukonen, V., Paasonen, P., Kurtén, T., Kulmala, M. & Vehkamäki, H. (2012). Atmospheric Cluster Dynamics Code: a flexible method for solution of the birth-death equations. *Atmos Chem Phys*, 12, 2345-2355, doi:10.5194/acp-12-2345-2012.

McMurry, P. H. & Grosjean, D. (1985). Gas and aerosol wall losses in Teflon film smog chambers. *Environmental Science & Technology*, 19, 1176-1182, doi:10.1021/es00142a006.

Myllys, N., Kubečka, J., Besel, V., Alfaouri, D., Olenius, T., Smith, J. N. & Passananti, M. (2019). Role of base strength, cluster structure and charge in sulfuric-acid-driven particle formation. *Atmos Chem Phys*, 19, 9753-9768, doi:10.5194/acp-19-9753-2019.

Olenius, T., Kupiainen-Määttä, O., Ortega, I. K., Kurtén, T. & Vehkamäki, H. (2013). Free energy barrier in the growth of sulfuric acid–ammonia and sulfuric acid–dimethylamine clusters. *J. Chem. Phys.*, 139, 084312, doi:10.1063/1.4819024.

Ortega, I. K., Kupiainen, O., Kurtén, T., Olenius, T., Wilkman, O., McGrath, M. J., Loukonen, V. & Vehkamäki, H. (2012). From quantum chemical formation free energies to evaporation rates. *Atmos Chem Phys*, 12, 225-235, doi:10.5194/acp-12-225-2012.

Pankow, J. F. (1994). An absorption model of gas/particle partitioning of organic compounds in the atmosphere. *Atmos. Environ.*, 28, 185-188, doi:10.1016/1352-2310(94)90093-0.

Pathak, R. K., Stanier, C. O., Donahue, N. M. & Pandis, S. N. (2007). Ozonolysis of α-pinene at atmospherically relevant concentrations: Temperature dependence of aerosol mass fractions (yields). *J Geophys Res*, 112, doi:10.1029/2006jd007436.

Pichelstorfer, L. & Hofmann, W. (2015). Modeling aerosol dynamics of cigarette smoke in a denuder tube. *J Aerosol Sci*, 88, 72-89, doi:10.1016/j.jaerosci.2015.05.009.

Pichelstorfer, L., Winkler-Heil, R., Boy, M. & Hofmann, W. (2021). Aerosol dynamics simulations of the anatomical variability of e-cigarette particle and vapor deposition in a stochastic lung. *J Aerosol Sci*, 158, 105706, doi:10.1016/j.jaerosci.2020.105706.

Pierce, J. R., Engelhart, G. J., Hildebrandt, L., Weitkamp, E. A., Pathak, R. K., Donahue, N. M., Robinson, A. L., Adams, P. J. & Pandis, S. N. (2008). Constraining Particle Evolution from Wall Losses, Coagulation, and Condensation-Evaporation in Smog-Chamber Experiments: Optimal Estimation Based on Size Distribution Measurements. *Aerosol Sci Tech*, 42, 1001-1015, doi:10.1080/02786820802389251.

Roldin, P., Ehn, M., Kurtén, T., Olenius, T., Rissanen, M. P., Sarnela, N., Elm, J., Rantala, P., Hao, L., Hyttinen, N., Heikkinen, L., Worsnop, D. R., Pichelstorfer, L., Xavier, C., Clusius, P., Öström, E., Petäjä, T., Kulmala, M., Vehkamäki, H., Virtanen, A., Riipinen, I. & Boy, M. (2019). The role of highly oxygenated organic molecules in the Boreal aerosol-cloud-climate system. *Nat Commun*, 10, doi:10.1038/s41467-019-12338-8.

Roldin, P., Eriksson, A. C., Nordin, E. Z., Hermansson, E., Mogensen, D., Rusanen, A., Boy, M., Swietlicki, E., Svenningsson, B., Zelenyuk, A. & Pagels, J. (2014). Modelling non-equilibrium secondary organic aerosol formation and evaporation with the aerosol dynamics, gas- and particle-phase chemistry kinetic multilayer model ADCHAM. *Atmos Chem Phys*, 14, 7953-7993, doi:10.5194/acp-14-7953-2014.

Roldin, P., Swietlicki, E., Schurgers, G., Arneth, A., Lehtinen, K. E. J., Boy, M. & Kulmala, M. (2011). Development and evaluation of the aerosol dynamics and gas phase chemistry model ADCHEM. *Atmospheric chemistry and physics*, 11, 5867-5896, doi:.

Rose, C., Chaumerliac, N., Deguillaume, L., Perroux, H., Mouchel-Vallon, C., Leriche, M., Patryl, L. & Armand, P. (2018). Modeling the partitioning of organic chemical species in cloud phases with CLEPS (1.1). *Atmos Chem Phys*, 18, 2225-2242, doi:10.5194/acp-18-2225-2018.

Shiraiwa, M., Pfrang, C. & Pöschl, U. (2010). Kinetic multi-layer model of aerosol surface and bulk chemistry (KM-SUB): the influence of interfacial transport and bulk diffusion on the oxidation of oleic acid by ozone. *Atmos Chem Phys*, 10, 3673-3691, doi:10.5194/acp-10-3673-2010.

Smith, J. N., Draper, D. C., Chee, S., Dam, M., Glicker, H., Myers, D., Thomas, A. E., Lawler, M. J. & Myllys, N. (2021). Atmospheric clusters to nanoparticles: Recent progress and challenges in closing the gap in chemical composition. *J Aerosol Sci*, 153, 105733, doi:10.1016/j.jaerosci.2020.105733.

Vignati, E., Wilson, J. & Stier, P. (2004). M7: An efficient size-resolved aerosol microphysics module for large-scale aerosol transport models. *Journal of Geophysical Research: Atmospheres*, 109, n/a-n/a, doi:10.1029/2003jd004485.

Xavier, C., Baykara, M., de Jonge, R. W., Altstädter, B., Clusius, P., Vakkari, V., Thakur, R., Beck, L., Becagli, S., Severi, M., Traversi, R., Wehner, B., Sipilä, M., Kulmala, M., Boy, M. & Roldin, P. (2022). Secondary aerosol formation in marine Arctic environments: A model measurement comparison at Ny-Å#lesund#. , , doi:10.5194/acp-2022-200.

Xavier, C., Rusanen, A., Zhou, P., Dean, C., Pichelstorfer, L., Roldin, P. & Boy, M. (2019). Aerosol mass yields of selected biogenic volatile organic compounds – a theoretical study with nearly explicit gas-phase chemistry. *Atmos Chem Phys*, 19, 13741-13758, doi:10.5194/acp-19-13741-2019.

Yli-Juuti, T., Barsanti, K., Ruiz, L. H., Kieloaho, A.-J., Makkonen, U., Petäjä, T., Ruuskanen, T., Kulmala, M. & Riipinen, I. (2013). Model for acid-base chemistry in nanoparticle growth (MABNAG). *Atmos Chem Phys*, 13, 12507-12524, doi:10.5194/acp-13-12507-2013.

Zhang, Q., Jimenez, J. L., Canagaratna, M. R., Allan, J. D., Coe, H., Ulbrich, I., Alfarra, M. R., Takami, A., Middlebrook, A. M., Sun, Y. L., Dzepina, K., Dunlea, E., Docherty, K., DeCarlo, P. F., Salcedo, D., Onasch, T., Jayne, J. T., Miyoshi, T., Shimono, A., Hatakeyama, S., Takegawa, N., Kondo, Y., Schneider, J., Drewnick, F., Borrmann, S., Weimer, S., Demerjian, K., Williams, P., Bower, K., Bahreini, R., Cottrell, L., Griffin, R. J., Rautiainen, J., Sun, J. Y., Zhang, Y. M. & Worsnop, D. R. (2007). Ubiquity and dominance of oxygenated species in organic aerosols in anthropogenically-influenced Northern Hemisphere midlatitudes. *Geophys Res Lett*, 34, n/a-n/a, doi:10.1029/2007gl029979.

Zhang, X., Cappa, C. D., Jathar, S. H., McVay, R. C., Ensberg, J. J., Kleeman, M. J. & Seinfeld, J. H. (2014). Influence of vapor wall loss in laboratory chambers on yields of secondary organic aerosol. *Proc. Natl. Acad. Sci.*, 111, 5802-5807, doi:10.1073/pnas.1404727111.

Zhang, Y., Seigneur, C., Seinfeld, J. H., Jacobson, M. Z. & Binkowski, F. S. (1999). Simulation of Aerosol Dynamics: A Comparative Review of Algorithms Used in Air Quality Models. *Aerosol Sci Tech*, 31, 487-514, doi:10.1080/027868299304039.